# SARS-CoV-2 Omicron is an immune escape variant with an altered cell entry pathway

Brian J. Willett [1,134 ✉], Joe Grove [1,134 ✉], Oscar A. MacLean[1,134], Craig Wilkie [2,134], Giuditta De Lorenzo [1,134], Wilhelm Furnon [1,134], Diego Cantoni[1,134], Sam Scott [1,134], Nicola Logan[1,134], Shirin Ashraf [1,134], Maria Manali[1], Agnieszka Szemiel[1], Vanessa Cowton [1], Elen Vink [1], William T. Harvey[1], Chris Davis[1], Patawee Asamaphan[1], Katherine Smollett[1], Lily Tong [1], Richard Orton[1], Joseph Hughes [1], Poppy Holland [3], Vanessa Silva[3], David J. Pascall [4], Kathryn Puxty [3], Ana da Silva Filipe [1], Gonzalo Yebra [5], Sharif Shaaban[5], Matthew T. G. Holden[5,6], Rute Maria Pinto[1], Rory Gunson[3], Kate Templeton[7], Pablo R. Murcia [1], Arvind H. Patel [1], Paul Klenerman[8], Susanna Dunachie [8], PITCH Consortium*, The COVID-19 Genomics UK (COG-UK) Consortium*, John Haughney[3,135], David L. Robertson [1,135], Massimo Palmarini[1,135], Surajit Ray [2,135] and Emma C. Thomson [1,3,9,135 ✉]

Vaccines based on the spike protein of SARS-CoV-2 are a cornerstone of the public health response to COVID-19. The emergence of hypermutated, increasingly transmissible variants of concern (VOCs) threaten this strategy. Omicron (B.1.1.529), the fifth VOC to be described, harbours multiple amino acid mutations in spike, half of which lie within the receptor-binding domain. Here we demonstrate substantial evasion of neutralization by Omicron BA.1 and BA.2 variants in vitro using sera from individuals vaccinated with ChAdOx1, BNT162b2 and mRNA-1273. These data were mirrored by a substantial reduction in real-world vaccine effectiveness that was partially restored by booster vaccination. The Omicron variants BA.1 and BA.2 did not induce cell syncytia in vitro and favoured a TMPRSS2-independent endosomal entry pathway, these phenotypes mapping to distinct regions of the spike protein. Impaired cell fusion was determined by the receptor-binding domain, while endosomal entry mapped to the S2 domain. Such marked changes in antigenicity and replicative biology may underlie the rapid global spread and altered pathogenicity of the Omicron variant.

Vaccination against SARS-CoV-2 is based primarily on vaccines that induce immunity to the spike glycoprotein. These vaccines have become the cornerstone of the global public health response to SARS-CoV-2[1]. However, their effectiveness is now being threatened by the emergence of variants of concern (VOC) displaying enhanced transmissibility and evasion of host immunity[2]. Of the five VOCs that have emerged, the Beta (B.1.351) and Gamma (P.1) variants were primarily associated with immune evasion, spreading internationally but never dominating globally. In contrast, the Alpha (B.1.1.7) and Delta (B.1.617.2) VOCs spread globally and were responsible for notable waves of infections and an increase in reproduction number ($R_0$). The Alpha and Delta variants harbour mutations within the polybasic cleavage site in spike (an H681 in Alpha and R681 in Delta) that enhance cleavage by furin—changes that are associated with enhanced cell entry and may contribute to increased transmissibility. While the Alpha variant spread rapidly, it was in turn replaced by the Delta variant that combined augmented transmissibility with immune evasion[2–5].

Omicron (lineage B.1.1.529) is the fifth variant to be named as a VOC by the World Health Organization (WHO) and was first detected in mid-November 2021 in Botswana, South Africa[6] and in quarantined travellers in Hong Kong[7]. It has since split into three divergent sub-lineages (BA.1, BA.2 and BA.3) of which BA.1 and BA.2 now dominate worldwide.

Emerging data indicate that the Omicron variant evades neutralization by sera obtained from people vaccinated with 1 or 2 doses of vaccine, especially when antibody titres are waning. Indicative studies have shown that 3 doses of spike-based vaccines may provide only partial protection from infection with this variant. Immune evasion by Omicron may have contributed to the extremely high transmission rates in countries with high vaccination rates or natural immunity ($R_0$ of 3–5 in the UK)[8–18].

In this study, we investigate the antigenic and biological properties of the Omicron variant that might underlie immune evasion and increased transmission of the virus using in vitro assays and real-life population data.

[1]MRC-University of Glasgow Centre for Virus Research, University of Glasgow, Glasgow, UK. [2]School of Mathematics & Statistics, University of Glasgow, Glasgow, UK. [3]NHS Greater Glasgow & Clyde, Glasgow, UK. [4]MRC Biostatistics Unit, University of Cambridge, Cambridge, UK. [5]Public Health Scotland, Glasgow, UK. [6]School of Medicine, University of St Andrews, St Andrews, UK. [7]NHS Lothian, Edinburgh, UK. [8]University of Oxford, Oxford, UK. [9]London School of Hygiene and Tropical Medicine, London, UK. [134]These authors contributed equally: Brian J. Willett, Joe Grove, Oscar A. MacLean, Craig Wilkie, Giuditta De Lorenzo, Wilhelm Furnon, Diego Cantoni, Sam Scott, Nicola Logan, Shirin Ashraf. [135]These authors jointly supervised this work: John Haughney, David L. Robertson, Massimo Palmarini, Surajit Ray, Emma C. Thomson. *Lists of members and their affiliations appear at the end of the paper. ✉e-mail: brian.willett@glasgow.ac.uk; joe.grove@glasgow.ac.uk; emma.thomson@glasgow.ac.uk

## Results

**Omicron displays substantial changes within spike predicted to affect antigenicity and furin cleavage.** Omicron is characterized by multiple changes within the receptor-binding domain (RBD) of the spike glycoprotein—regions targeted by class 1, 2 and 3 RBD-directed antibodies—and within the N-terminal domain (NTD) supersite (Fig. 1a). Within the spike protein, BA.1 and BA.2 sub-lineages share 21 amino acid mutations with 12 distinct mutations in BA1 and 6 in BA.2. BA.2 lacks the 69,70 deletion present in BA.1. The G339D, N440K, S477N, T478K, Q498R and N501Y mutations (present in BA.1 and BA.2) enhance binding of spike to the human ACE2 receptor, while combinations such as Q498R and N501Y may enhance ACE2 binding additively[19]. Deep mutational scanning (DMS) estimates at mutated sites are predictive of substantially reduced monoclonal and polyclonal antibody binding and altered binding to human ACE2 (Fig. 1b)[20]. Fourteen mutations in Omicron (K417N, G446S (BA.1), E484A, Q493R, G496S (BA.1), Q498R and to a lesser extent, G339D, S371L/F (BA.1/BA.2), S373P, N440K, S477N, T478K, N501Y and Y505H) may affect antibody binding on the basis of a calculated escape fraction (a quantitative measure of the extent to which a mutation reduces polyclonal antibody binding by DMS). Seven Omicron RBD mutations (K417N, G446S(BA.1), E484A, Q493R, G496S(BA.1), Q498R and N501Y) have been previously shown to be associated with decreased antibody binding, importantly falling in epitopes corresponding to three major classes of RBD-specific neutralizing antibodies (nAbs). The mutations present in spike also involve key structural epitopes targeted by several monoclonal antibodies in current clinical use. Of these, bamlanivimab, cilgavimab, casirivimab, etesevimab, imdevimab, regdanvimab and tixagevimab bind to the receptor binding motif, and neutralization of Omicron has been shown to be negligible or absent. In contrast, sotrovimab targets a conserved epitope common to SARS-CoV-1 and SARS-CoV-2 that is outside the receptor binding motif, and has only a small reduction (3×) in neutralization potency in BA.1[21–23]. N679K and P681H mutations at the furin cleavage site are predicted individually to increase furin cleavage, although the combination of these changes and an adjacent change (H655Y, also present in the Gamma VOC) in the vicinity of the cleavage site is unknown[24].

Omicron bears several deletions (amino acids 69–70, 143–145 and 211 in BA.1; amino acids 24–26 in BA.2) and an insertion in BA.1 at site 214 in the NTD of spike. The 69–70 deletion is also found in the Alpha and Eta (B.1.525) variants and is associated with enhanced fusogenicity and incorporation of cleaved spike into virions[25]. This 69–70 deletion is currently a useful proxy for estimates of BA.1 prevalence in the population by S-gene target failure (SGTF) using the TaqPath (Applied Biosystems) diagnostic assay. Deletions in the vicinity of amino acids 143–145 have been shown to affect a range of NTD-specific nAbs[26,27].

**Emergence of the Omicron variant in the UK.** Despite high vaccination rates and levels of natural immunity following previous exposure in the UK, Omicron has rapidly become dominant. The evolutionary relationships of SARS-CoV-2 variants at a global level are shown in Fig. 1c. The first cases of Omicron (BA.1) were detected in the UK on 27 and 28 November 2021 (2 in England and 6 in Scotland). The proportion of variants in Scotland between September 2021 and February 2022 is shown in Fig. 1d, highlighting the rise of BA.1, BA.1.1 and then more recently BA.2 in sequential waves.

**Neutralizing responses to Omicron (BA.1 and BA.2) are substantially reduced following double vaccination and partially restored following triple vaccination.** Levels of nAbs in patient sera correlate strongly with protection from infection[28–31], and reductions in neutralizing activity against the Alpha and Delta variants are consistent with a reduction in vaccine effectiveness[2–5,32]. To predict the effect of the mutations within the Omicron spike

glycoprotein on vaccine effectiveness, sera collected from healthy volunteers at more than 14 d post second-dose vaccination with either BNT162b2, ChAdOx1 or mRNA-1273 were sorted into three age-matched groups ($n=24$ per group, mean age 45 years). Sera were first screened by multiplex meso scale discovery electrochemiluminescence (MSD-ECL) assay for reactivity with SARS-CoV-2 antigens (Spike, RBD, NTD or nucleoprotein (N)). The antibody responses to RBD and NTD were significantly higher ($P<0.0001$) in the sera from individuals vaccinated with BNT162b2 or mRNA-1273 in comparison with the ChAdOx1 vaccinees (Fig. 2a and Supplementary Table 1). In contrast, antibody responses to endemic human coronaviruses (HCoVs) (Extended Data Fig. 1 and Supplementary Table 2) or influenza (Extended Data Fig. 2 and Supplementary Table 3) were similar, except for coronavirus OC43 where responses in BNT162b2 and ChAdOx1 vaccinees differed significantly, perhaps suggesting modulation (back-boosting) of pre-existing OC43 responses by BNT162b2 vaccination.

Next, the nAb responses against SARS-CoV-2 pseudotypes expressing the spike glycoprotein from either the dominant variant of the first wave, lineage B.1 (defined by the spike mutation D614G), or Omicron (BA.1) were compared (Fig. 2b). Vaccination with mRNA-1273 elicited the highest nAb titres (mean titre B.1 = 21,118, Omicron (BA.1) = 285) in comparison with those elicited by vaccination with either BNT162b2 (B.1 = 4,978, Omicron (BA.1) = 148.3) or ChAdOx1 (B.1 = 882.3, Omicron (BA.1) = 61.9). Neutralizing antibody titres against B.1 differed significantly among the three study groups. Activity against Omicron (BA.1) was markedly reduced in comparison with B.1, reduced by 33-fold for BNT162b2, 14-fold for ChAdOx1 and 74-fold for mRNA-1273 (Supplementary Table 4). While the fold change in neutralization was lowest in recipients of the ChAdOx1 vaccine and highest in recipients of the mRNA-1273 vaccine, absolute neutralization values were highest in mRNA-1273, followed by BNT162b2 and ChAdOx1. Neutralization was lowest in the ChAdOx1 group; however, it is important to note that this was given to older patients during early vaccine rollout in the UK, especially to vulnerable patients in nursing homes and was not recommended in young adults less than 40 years of age.

Next, samples were analysed from vaccine recipients at least 14 d post booster vaccination (third dose). Participants had been primed with two doses of either ChAdOx1 ($2.5\times10^8$ IU) or BNT162b2 (30 μg), followed by a third dose of either BNT162b2 (30 μg) or mRNA-1273 (50 μg). All sera reacted strongly to SARS-CoV-2 antigens by MSD-ECL, with no significant differences between the four groups (Fig. 2c and Supplementary Table 5). Antibody responses to HCoVs (Extended Data Fig. 3 and Supplementary Table 6) or influenza (Extended Data Fig. 4 and Supplementary Table 7) were similar, with the exception of influenza Michigan H1, where responses in ChAdOx1-primed and BNT162b2 or mRNA-1273-boosted groups differed significantly, probably reflecting co-administration of influenza booster vaccines during the booster campaign. Two vaccine recipients boosted with BNT162b2 displayed weak reactivity to nucleocapsid (Fig. 2c), suggesting previously undetected exposure to SARS-CoV-2. Sera from vaccine recipients primed with BNT162b2 and boosted with either BNT162b2 or mRNA-1273 displayed similar titres of nAb against B.1 to the samples collected post dose 2 (Fig. 2d). In contrast, vaccination of individuals primed with ChAdOx1 with a booster dose of either BNT162b2 or mRNA-1273 resulted in a marked increase in antibody titre (9.3-fold increase) against B.1 relative to the low titres after dose 2 (Fig. 2e and Supplementary Table 8). The marked increase in antibody titre in ChAdOx1-primed individuals (Extended Data Fig. 5) emphasizes the importance of the third dose booster in this population. Indeed, following boost with either BNT162b2 or mRNA-1273, anti-B.1 nAb titres in the ChAdOx1-primed group were not significantly different from those primed with BNT162b2 (Supplementary Table 8). NAb titres against Omicron (BA.1) were lower in both booster study groups and did

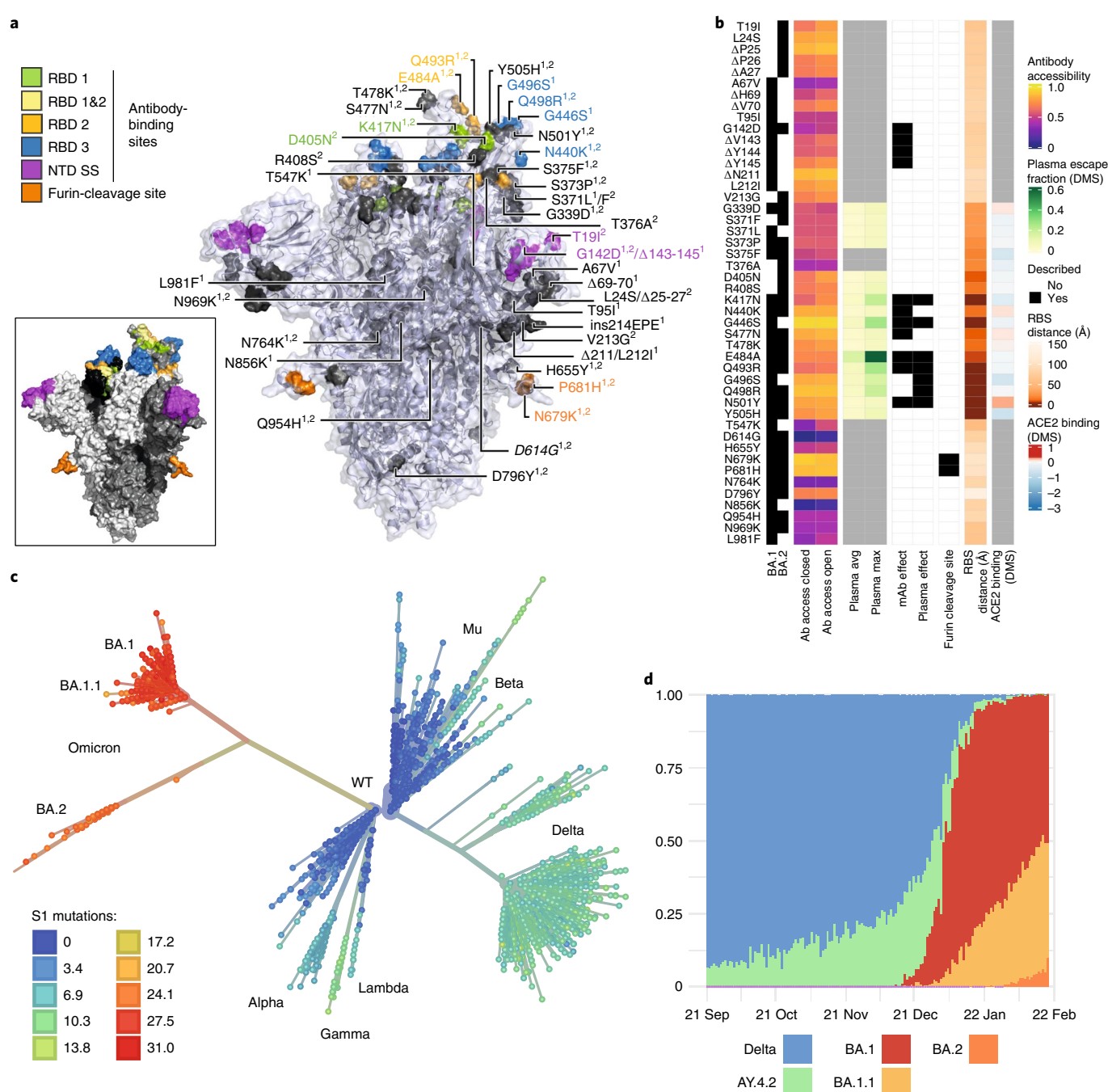

**Fig. 1 | Spike amino acid changes, phylogeny and emergence of the Omicron variant. a,** Spike homotrimer in open conformation with locations of Omicron substitutions, deletions (Δ) or insertions (ins) present in the lineages BA.1 and BA.2 (superscripts 1,2). Mutated residues highlighted as spheres with opaque surface representation. Residues impacting RBD-specific antibodies of classes 1, 2 or 3[91]; belonging to the NTD antibody supersite (NTD SS)[26]; or comprising the furin-cleavage site, are coloured, the remainder in grey. Inset shows the wider extent of these sites, with remaining areas of the protein shaded to show the three monomers. Mutations are annotated on the monomer with an 'up' receptor-binding domain with D614G (italicized), which is shared by common descent by all lineage B.1 descendants. Visualization uses a complete spike model[84] based on a partial cryo-EM structure (RCSB Protein Data Bank (PDB) ID: 6VSB[92]). **b,** Heat maps showing properties of amino acid residues in the Omicron variants BA.1 and BA.2. Structure-based epitope scores[87] for residues in the spike structure in closed and open conformations are shown. For RBD residues, DMS studies show the escape fraction (quantitative measure of the extent to which a mutation reduced polyclonal antibody binding) for each mutant-averaged ('plasma average') and most sensitive plasma ('plasma max')[20]. Each mutation is classified as having mutations affecting neutralization by either monoclonal antibodies (mAbs)[27,85,93,94,95] or antibodies in convalescent plasma (infected or vaccinated[20,85,95,96]). Membership of the furin cleavage site is shown. Distance to ACE2-contacting residues forming the receptor-binding site (RBS) is shown ('RBS' is residues with an atom <4 Å of an ACE2 atom in the structure of RBD bound to ACE2) (RCSB PDB ID: 6M0J[86]). ACE2 binding scores represent the binding constant ($\Delta\log_{10} K_D$) relative to the wild-type reference amino acid from DMS experiments[97]. **c,** Inferred evolutionary relationships of SARS-CoV-2 from NextStrain (https://nextstrain.org/ncov/gisaid/global), with the VOCs labelled. Tree tip colours correspond to the number of mutations causing spike amino acid substitutions relative to the Wuhan-Hu-1 (lineage B) sequence. **d,** The proportion of genome sequences from Scotland sampled between 1 September 2021 and 29 January 2022 is shown: 'Delta' includes all B.1.617.2 and AY assigned sequences that are not in the Delta sub-lineage AY.4.2 (defined by spike mutations Y145H and A222V), and BA.1, BA.1.1 and BA.2 are Omicron sub-lineages.

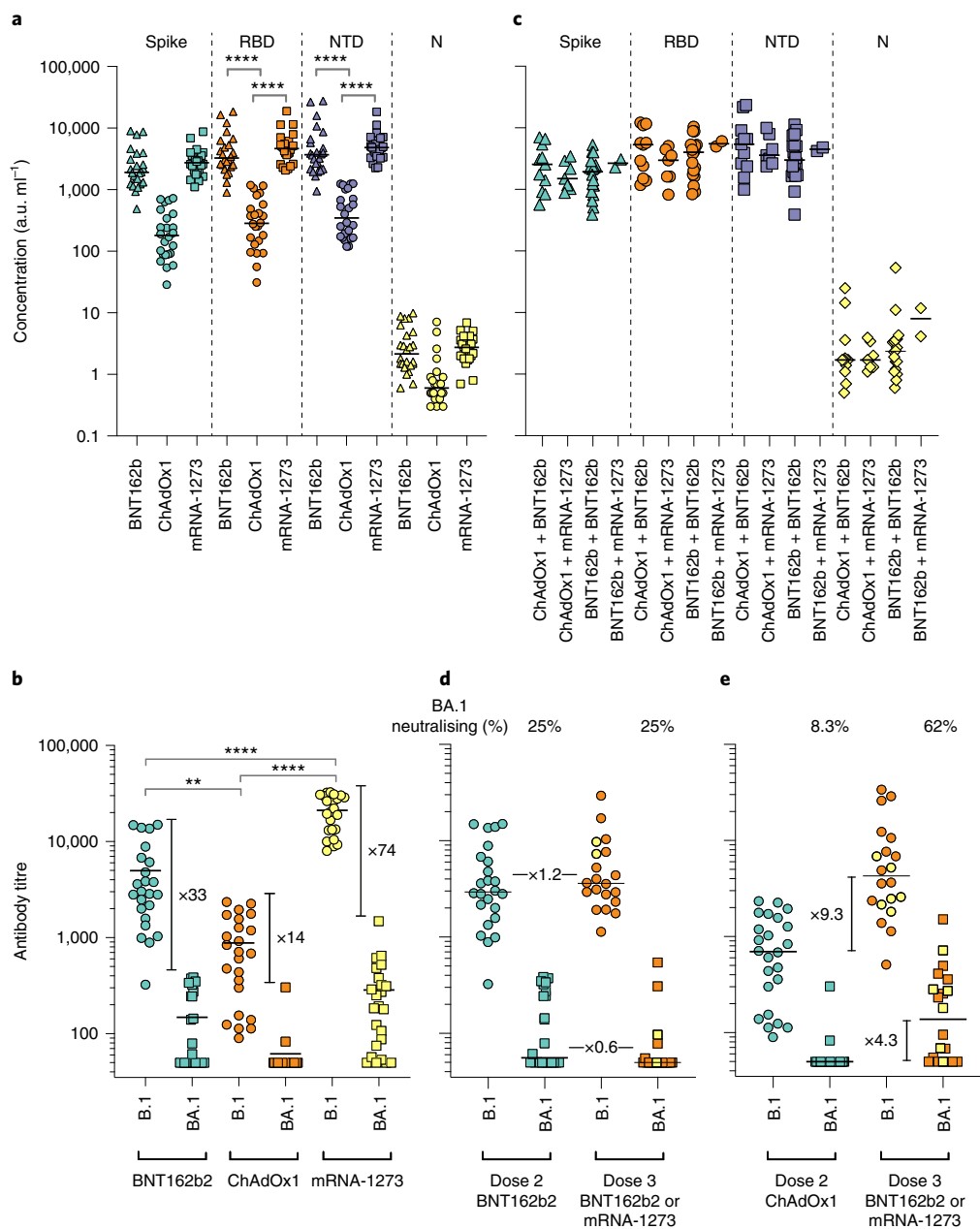

**Fig. 2 | Antibody responses elicited by SARS-CoV-2 vaccination. a,b**, Antibody responses were studied in three groups of individuals ($n = 24$ per group) receiving primary vaccination with either BNT162b2, ChAdOx1 or mRNA-1273 by MSD-ECL assay (**a**) or pseudotype-based neutralization assay (**b**). **a**, Responses were measured against full-length spike glycoprotein (Spike), RBD, NTD and N, and are expressed as arbitrary units (a.u. ml$^{-1}$). Horizontal bar represents group mean; between group comparisons by ordinary one-way analysis of variance (ANOVA), Tukey's multiple comparisons test; ****$P < 0.0001$. **b**, NAb responses were quantified against B.1 lineage (D614G) or Omicron (BA.1) spike glycoprotein bearing HIV (SARS-CoV-2) pseudotypes. Each point represents the mean of three replicates, horizontal bar represents the group mean; between group comparisons by ordinary one-way ANOVA, Tukey's multiple comparisons test; **$P = 0.0075$, ****$P < 0.0001$. **c–e**, To assess the effect of booster vaccines, antibody responses were studied in two groups of individuals primed with two doses of either BNT162b2 or ChAdOx1 and boosted with either BNT162b2 or mRNA-1273. Reactivity against SARS-CoV-2 antigens was measured by MSD-ECL assay (**c**), while neutralizing activity (**d,e**) was measured using HIV (SARS-CoV-2) pseudotypes, as above. Yellow data points represent those boosted with mRNA-1273, all others received BNT162b2. In **d** and **e**, 'BA.1 neutralizing (%)' refers to the proportion of serum samples that displayed neutralizing activity against Omicron pseudotypes.

not differ significantly (Supplementary Table 8). However, absolute numbers displaying measurable Omicron neutralizing activity were higher in the ChAdOx1-primed group (13/21, 62%) compared with the BNT162p2 primed group (5/20, 25%) (Fig. 2d,e).

Since the arrival of Omicron in the UK, three distinct lineages have emerged—BA.1, BA.1.1 (BA.1 + R346K) and BA.2. Therefore, we asked whether the cross-neutralizing antibodies elicited by

third dose booster vaccination retained activity against BA.1.1 and BA.2 (Fig. 3). While neutralization of Omicron BA.1, BA.1.1 and BA.2 was significantly lower than that of the first wave variant B.1 ($P < 0.0001$), neutralization of BA.1, BA.1.1 and BA.2 did not differ significantly from each other (Fig. 3a). Administration of a third dose boosted neutralization of each Omicron variant (Fig. 3b); however, individuals varied in their responses to boosting (Fig. 3c).

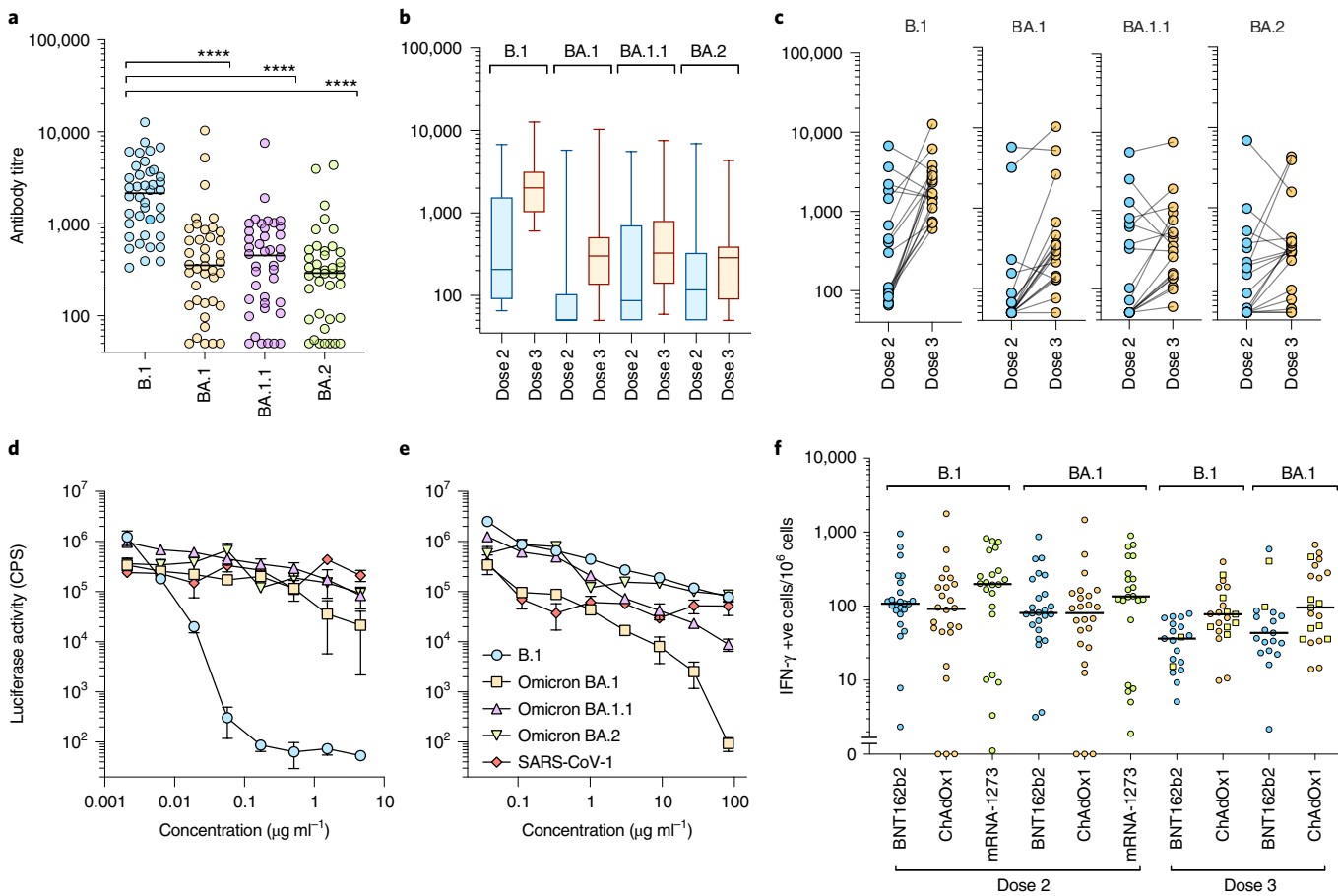

**Fig. 3 | Neutralization of Omicron variants by third dose sera. a–c**, Sensitivity of B.1, Omicron BA.1, BA.1.1 and BA.2 to neutralization by sera elicited following third dose booster vaccination. **a**, All third dose sera irrespective of vaccine type. Each point represents the mean of three technical replicates, bar represents the mean. Group means compared by one-way ANOVA, Tukey's multiple comparisons test; ****$P < 0.0001$. **b**, Neutralizing activity between matched second and third dose vaccine sera. Box and whisker plots (median and range). **c**, Comparison of matched sera after second or third dose vaccination. Each point represents the mean of three technical replicates ± s.e.m. **d,e**, Activity of Ronapreve (casirivimab and imdevimab) (**d**) and Xevudy (sotrovimab) (**e**) against pseudotypes bearing B.1, BA.1, BA.1.1, BA.2 and SARS-CoV-1 spike glycoproteins. Each point represents the mean of three technical replicates ± s.e.m. CPS, counts per second. **f**, Interferon-γ production by PBMC from vaccinated individuals in response to B.1 or BA.1 S-derived peptides. Dose 3 boost with BNT162b2 (circles) or mRNA-1273 (squares).

Where neutralization was low after dose 2, in general, the third dose increased neutralization. However, where neutralization was high after dose 2, in some individuals, the third dose boost had little effect or a decrease in titre was noted.

The antigenic divergence of Omicron from earlier virus variants has raised concerns regarding the effectiveness of monoclonal antibody-based therapies. Each of the variants tested (BA.1, BA.1.1, BA.2) resisted neutralization with Ronapreve (combined casirivimab and imdevimab, Fig. 3d), while Ronapreve neutralized B.1 effectively. In comparison, Xevudy (sotrovimab) retained neutralizing activity against BA.1 and to a lesser extent BA.1.1 (Fig. 3e). Activity against BA.2 was reduced in comparison, mirroring that against B.1.

In contrast to neutralizing antibody responses, T-cell responses measured by IFN-γ ELISpot[33] to Omicron (BA.1) versus B.1-derived peptides were similar, in agreement with minimal variation at key antigenic sites (Fig. 3f).

**Vaccine effectiveness against the Omicron variant is reduced compared with the Delta variant.** A logistic additive model with a test negative case control design was used to estimate relative vaccine effectiveness against becoming a confirmed case with Delta (5,689 cases) and/or Omicron (17,699 cases) in a population of

1.2 million people in the largest health board in Scotland, NHS GG&C, between 6 and 26 December 2021. Demographic data are shown in Supplementary Table 9. The timings of first, second and third doses of vaccination are shown in Fig. 4a and the occurrence of sequenced/confirmed infections with different variants in vaccine recipients over time is shown in Fig. 4b. Infection status for Omicron and Delta was modelled by number and product type of vaccine doses, previous infection status, sex, Scottish Index of Multiple Deprivation (SIMD) vigintile, and age (to control for demographic bias). Immunosuppressed individuals were removed from the analysis to ensure case-positivity could be attributed to vaccine escape rather than an inability to mount a vaccine response, with immunosuppression status derived from a list of those in GG&C either shielding due to immunosuppression or being given immunosuppressant drugs. Age and SIMD vigintile were each modelled as single smooth effects using thin plate regression splines[34]. Vaccine product, vaccine dose number and previous infection status were combined into a single categorical variable with a base level of unvaccinated and not previously infected.

The protection from vaccine-acquired and infection-acquired immunity was estimated as being markedly reduced against Omicron compared with Delta. Estimates of vaccine effectiveness

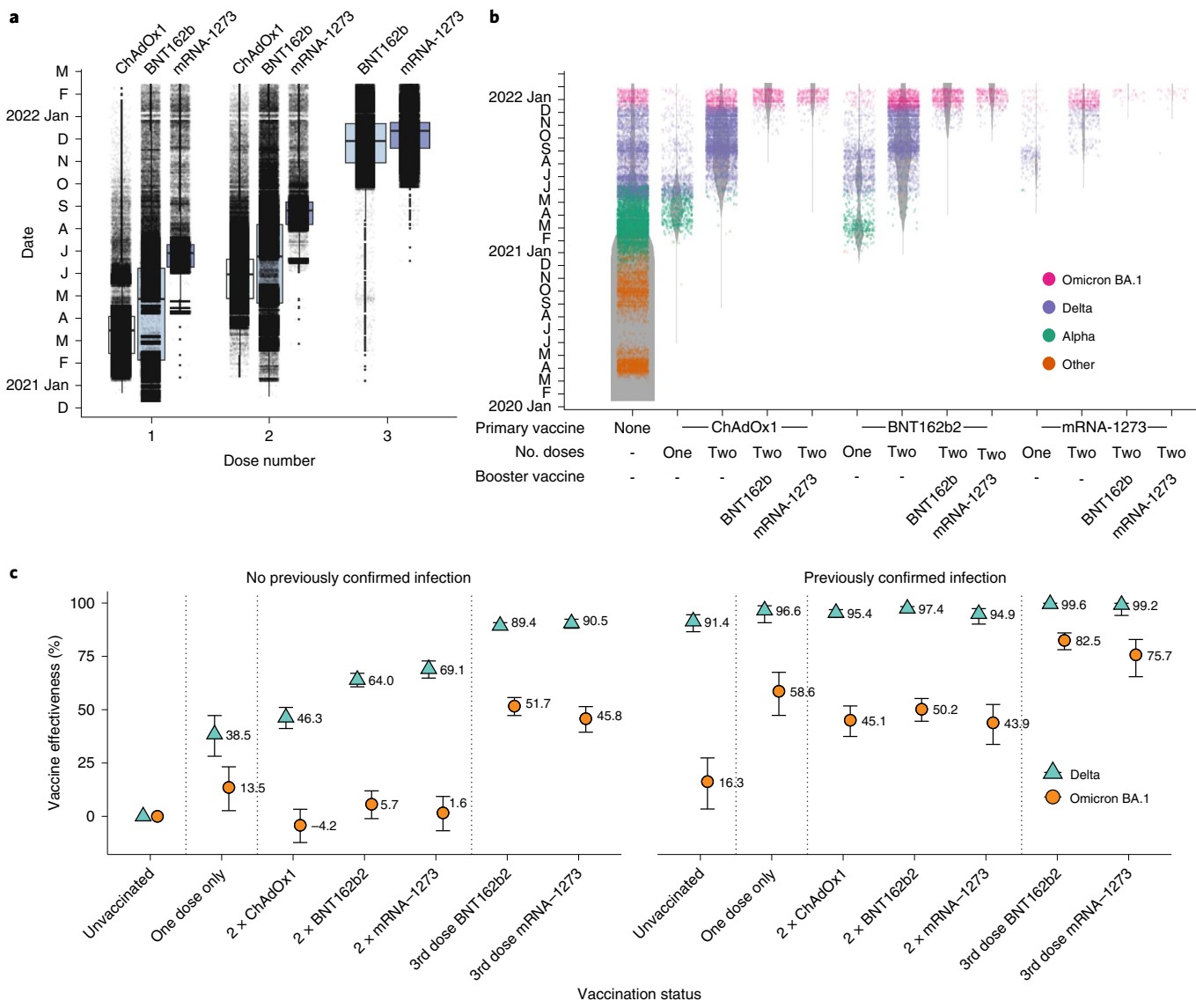

**Fig. 4 | Vaccine deployment and vaccine effectiveness. a**, Boxplots of date of first, second and third administered vaccine dose by vaccine product for the population of NHS GG&C aged 18 years and older. The box limits are the quartiles and the centre line is the median, with whisker length of 1.5 times the interquartile range. Outliers are shown as dots outside the whisker range. Data points are overlaid as a dot plot with points shown as black dots, with a random jitter along the x axis applied for visual clarity. **b**, Denominator (violin) plot showing populations of test positive and test negative cohorts in NHS GG&C, with the widths of the grey bands representing the populations in each group at each time point. VOC classifications of sequenced cases are overlaid as a dot plot, with points coloured by their VOC and a random jitter applied along the x axis for visual clarity. **c**, Error bar plot of estimated vaccine effectiveness against testing positive for Delta and Omicron BA.1 SARS-CoV-2 infection in the population of over 18 years in NHS GG&C who were tested between 6 and 26 December 2021. The points and corresponding numbers represent the estimated vaccine effectiveness (%) for each group and for each variant, with the error bars representing 95% confidence intervals.

(at least 14 d post dose) for those with no previous confirmed SARS-CoV-2 infection were 46.3% for two-dose primary courses of ChAdOx1 against Delta, 64.0% for two doses of BNT162b2 and 69.1% for 2 doses of mRNA-1273, but the corresponding estimates for Omicron were not significantly different from zero (Fig. 4c). These low estimates are probably due to the waning effects of vaccination over time (Extended Data Fig. 6). The responses increased significantly following a third booster dose of BNT162b2 or mRNA-1273, without previous confirmed infection, against Delta (89.4% and 90.5%, respectively), and against Omicron (51.7% and 45.8%, respectively). These estimates are in keeping with those reported recently against symptomatic infection in England where vaccine effectiveness was estimated as 71.4% and 75.5% for ChAdOx1 and

BNT162b2 primary course recipients, respectively, after boosting with BNT162b2[9].

Next, we estimated the protective effect of previous natural infection (at least 90 d previously) and its interaction with the protective effect from vaccination. Given that we can only estimate effectiveness in those who survived their first infection, it is possible that we are analysing an above-average immunological cohort. Acquired immunity directed against natural infection may be broader in nature and may wane more slowly than that induced by vaccines[35–37]. We found a significantly reduced protective effect of previous infection against Omicron compared with Delta for all vaccination groups and for those who were unvaccinated. We estimated a high protective effect of previous infection against subsequent infection

with Delta of 91.4% for those unvaccinated, rising to 99.6% and 99.2% for those previously infected, and then boosted with a third dose of BNT162b2 and mRNA-1273, respectively. We estimated a protective effect of previous infection against subsequent infection with Omicron of only 16.3% for those unvaccinated, but this increased significantly for those vaccinated with one or two doses of any vaccine, rising to 82.5% and 75.5% for those boosted with a third dose of BNT162b2 and mRNA-1272, respectively. Collectively, these results emphasize the importance of booster vaccines, irrespective of previous history of infection. Further, vaccine-mediated protection against severe disease will probably be more durable than that against detected infection[38].

**Absence of syncytia in Omicron-infected cells.** Our data demonstrate that antigenic change in Omicron permits evasion of vaccine-induced immunity; however, the constellation of spike mutations in Omicron suggests that functional change may also contribute to its rapid transmission (Fig. 1a). Therefore, we investigated the virological properties of live Omicron (BA.1) isolated from a patient sample. SARS-CoV-2 particles can achieve membrane fusion at the cell surface following proteolytic activation of spike by the plasma membrane protease TMPRSS2. This property also permits spike-mediated fusion of SARS-CoV-2-infected cells with adjacent cells resulting in syncytia[39]—a feature that has been associated with severe disease[40]. Moreover, the Delta variant has been shown to exhibit enhanced fusion compared with the Alpha and Beta variants[41].

A split green fluorescent protein (GFP) cell–cell fusion system[42] was used to quantify syncytia formation by Omicron, Delta and first wave lineage B.1 virus (Fig. 5a). Cells expressing split GFP were infected with B.1, Delta or Omicron BA.1 and the levels of the reconstituted GFP signal following cell–cell fusion were determined in real time (Fig. 5c). In addition, infected cells were probed by indirect immunofluorescence assay to assess viral replication by the detection of the viral nucleocapsid protein (Fig. 5b). The Delta variant exhibited the highest levels of cell fusion, followed by B.1. In contrast, Omicron BA.1 failed to form syncytia. This failure was not due to lack of infection as immunofluorescent detection of nucleocapsid protein confirmed viral replication by Omicron BA.1, B.1 and Delta[18]. Moreover, comparison of clinical isolates (as in Fig. 5b,c) and reverse genetics of live viruses in which the Delta or Omicron BA.1 spike was presented in the wild-type (WT) lineage B background, exhibited equivalent fusion activity (Extended data Fig. 7), demonstrating that the Omicron fusion defect is entirely attributable to changes within the spike protein.

**Reduced replication kinetics of Omicron BA.1 in lung epithelial cells.** The replication of Omicron BA.1, Delta and B.1 was compared in Calu-3, a human lung epithelial cell line. B.1 and Delta displayed comparable replication kinetics over a period of 72 h, with visible cytopathic effect (CPE) between 48–72 h post infection (Fig. 5d). In contrast, the titres of Omicron were at least an order of magnitude lower at each time point compared with B.1 and Delta. These observations are consistent with attenuated replication of Omicron BA.1 in lower respiratory tissues as recently reported[18,43].

**Omicron spike has switched entry route preference.** Entry of SARS-CoV-2 and related coronaviruses can proceed via two routes[44]: cell surface fusion following proteolysis by TMPRSS2, as described above ('route 1', Fig. 5e), or fusion from the endosome after endocytosis and activation by the endosomal proteases Cathepsin B or L ('route 2', Fig. 5e). The ability of SARS-CoV-2 to achieve cell surface fusion is thought to be dependent on its S1/S2 polybasic cleavage site; this is absent from most closely related sarbecoviruses, which are confined to endosomal fusion[45–47]. Given the reduced fusogenicity and replication kinetics of Omicron BA.1, HIV pseudotypes were

used to evaluate entry route preference. Wild-type lineage B (WT (B)), Alpha, Delta and Omicron (BA.1 and BA.2) spikes were examined, while Pangolin CoV (Guangdong isolate) spike was included as a control. Pangolin CoV spike exhibits high affinity interactions with human ACE2 but lacks a polybasic cleavage site and, therefore, enters via the endosome only[48–51].

Calu-3 cells support cell surface (route 1) fusion predominantly, owing to their high endogenous expression of TMPRSS2[46,52]. In these cells, Delta yielded the highest infection, being ~4-fold higher than that in Omicron BA.1 (Fig. 5f). Pangolin CoV infection was low, indicating that Calu-3 cells did not support robust endosomal entry. In contrast, human embryonic kidney (HEK) cells only supported endosomal entry and in these cells Pangolin CoV had high infection. Notably, Omicron BA.1 also achieved high infection in HEK cells, producing ~10-fold greater signal than Delta. This suggested that Omicron BA.1, like Pangolin CoV, was optimized for endosomal entry. All pseudotypes exhibited robust infection in A549-ACE2-TMPRSS2, where both entry routes are available[53,54].

Entry pathway preference was further investigated using protease inhibitors targeting either TMPRSS2 (Camostat) or cathepsins (E64d)[45]. In Calu-3 cells, all SARS-CoV-2 pseudotypes were inhibited by Camostat, whereas only Omicron (BA.1) exhibited E64d sensitivity, indicating that a component of infection occurs via endosomal entry (Fig. 5g). In HEK cells, all pseudotypes were inhibited by E64d, whereas Camostat was non-inhibitory, confirming that only endosomal entry was available in these cells. Inhibitor treatment in A549-ACE2-TMPRSS2 provided the clearest evidence of altered entry by Omicron. WT (lineage B), Alpha and Delta were potently inhibited by Camostat, but not by E64d. For Omicron BA.1 and Pangolin CoV, this pattern was reversed, suggesting a strong preference for endosomal fusion. This conclusion was supported by titration of either inhibitor in A549-ACE2-TMPRSS2 cells (Fig. 5h). Furthermore, live virus infection in the presence of protease inhibitors confirmed increased sensitivity of Omicron BA.1 to E64d compared with Delta (Extended Data Fig. 8). These data suggest that while Delta is optimized for fusion at the cell surface, Omicron BA.1 preferentially achieves entry through endosomal fusion. Immunoblotting of exogenously expressed spike indicated reduced proteolytic processing in Omicron BA.1 spike compared with ancestral WT (lineage B) or Delta spikes (Fig. 5I), consistent with reduced spike fusogenicity and altered entry mechanism. The related BA.2 Omicron variant exhibited a similar switch in entry pathway preference, as evidenced by pseudotype infection of HEK cells (Fig. 5j) and sensitivity to protease inhibitors (Extended Data Fig. 9a). The BA.2 spike also displayed a defect in syncytia formation equivalent to that of BA.1 (Extended Data Fig. 9b).

A switch in the Omicron entry pathway may explain its apparent preference for replication in upper airway tissues. Consistent with this, live Omicron BA.1 had a replicative advantage over Delta in human nasal epithelial cells (hNEC, Fig. 5k), in contrast to infection of Calu-3 cells, which are a lower airway-derived cell line (Fig. 5d). Immunoblotting of hNEC and Calu-3 cells demonstrated an opposing pattern of protease expression (Fig. 5l), with Calu-3 cells possessing high levels of TMPRSS2 (required for cell surface entry), whereas Cathepsin-L (required for endosomal entry) predominated in hNEC Cathepsin-L. This correlation suggests that entry pathway switching may determine tissue preference. When spike processing in infected hNEC and Calu-3 cells were compared, reduced cleavage of the Omicron BA.1 spike was observed, consistent with plasmid expressed protein (Fig. 5I). This characteristic may link mechanistically with reduced syncytia formation, hence we reasoned that the fusion defect (Fig. 5c) could be overcome by addition of exogenous protease to increase spike processing. Accordingly, trypsin was able to rescue Omicron BA.1 spike fusogenicity in a dose dependent manner, increasing it to a level equivalent to that in Delta spike-mediated fusion (Fig. 5m,n).

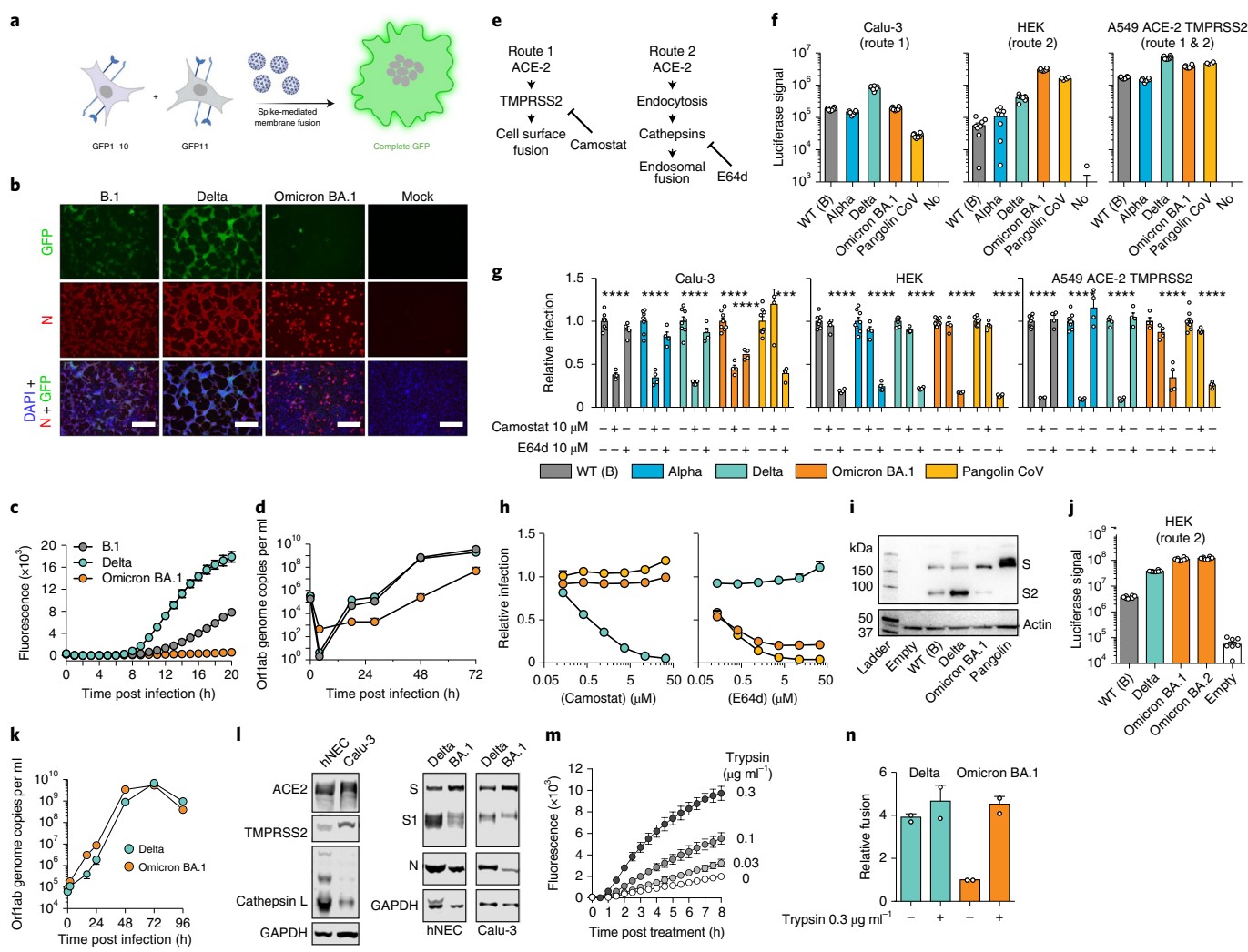

**Fig. 5 | Omicron exhibits reduced syncytia formation and has switched entry route. a**, Virus-induced fusion of co-cultured cells expressing split GFP (GFP-10 and GFP-11, respectively) reconstitutes fluorescence (image generated in BioRender, agreement SX23HJ6GY8SX23HJ6GY8). **b**, Co-cultured GFP-10 and GFP-11 A549-ACE2-TMPRSS2 cells were infected with B.1, Delta and Omicron/BA.1, photomicrographs taken at 22 h post infection: GFP, green; nucleocapsid (N), red; DAPI nuclei, blue; scale bars, 100 μm. **c**, Fluorescence over 20 h (n = 4 technical repeats, 3 independent experiments, 2 virus stocks). **d**, Calu-3 cell infection with B.1, Delta and Omicron/BA.1. Supernatants were assessed by RT-qPCR (n = 3 technical repeats, 2 independent experiments). **e**, SARS-CoV-2 entry routes: route 1 – fusion at the cell surface following processing by TMPRSS2; route 2 – fusion occurs post endocytosis after processing by cathepsins. Routes 1 and 2 are inhibited by Camostat and E64d, respectively. **f**, SARS-CoV-2 pseudotype infection of cell lines, mean luciferase values (1 experiment, n = 8 technical repeats, representative of 3 independent experiments). Route 1 predominates in Calu-3, route 2 in HEK, and both routes are supported by A549-ACE2-TMPRSS2. Control is Pangolin CoV (route 2 only), negative control is no glycoprotein pseudotypes (No). **g**, Relative pseudotype infection (compared to untreated) of 10 μM protease inhibitor-treated cells; mean of 4 biological repeats, ****P < 0.0001, one-way ANOVA, Dunnett's multiple comparisons test. **h**, Camostat (left) and E64d (right) titration against Delta, Omicron/BA.1 and Pangolin CoV in A549-ACE2-TMPRSS2; mean relative infection compared to untreated control (n = 2 biological repeats). **i**, Immunoblot of spike-expressing HEK lysates with anti-spike (S2) and anti-actin. **j**, HEK infection by Omicron/BA.2 pseudotype; mean luciferase values (1 experiment, n = 8 technical repeats, representative of 3 independent experiments). **k**, hNEC infection with Delta and Omicron/BA.1 clinical isolates, supernatants assessed by RT-qPCR; mean of 2 independent biological repeats. **l**, Immunoblots of uninfected (left) and infected (right) hNEC and Calu-3 lysates. Immunoblots probed for ACE2, TMPRSS2, Cathepsin-L, GAPDH, Spike S1 and N. **m**, Cell–cell fusion (as in **a**) in Omicron/BA.1-transfected cells treated with trypsin. Plot displays mean fluorescence (1 experiment, 3 technical repeats, representative of 2 independent experiments). **n**, Cell–cell fusion in Delta or Omicron/BA.1-transfected cells, ±0.3 μg ml⁻¹ trypsin. Data are relative to fusion by Omicron/BA.1 spike (−trypsin, 8 h), mean of 2 biological repeats. All error bars represent s.e.m.

These experiments indicate a fundamental change in the biology of Omicron (BA.1 and BA.2) spike. It has a reduced ability to form syncytia, most probably linked to changes in spike pre-processing at the S1/S2 boundary. Omicron spike is also optimized to preferential entry via the endosome, resulting in alterations in cellular tropism. This biological about-face may underpin the evident changes in Omicron transmission and pathogenesis.

**The Omicron spike phenotypes are conferred by distinct domains.** To investigate the determinants of Omicron spike biology, reciprocal domain swaps with the ancestral WT (lineage B) spike were performed (Fig. 6a). N-terminal domain (NTD) swaps included residues before position 319, receptor-binding domain (RBD) swaps included positions 320–576, and S2 swaps include positions downstream of 577 and, therefore, included mutations

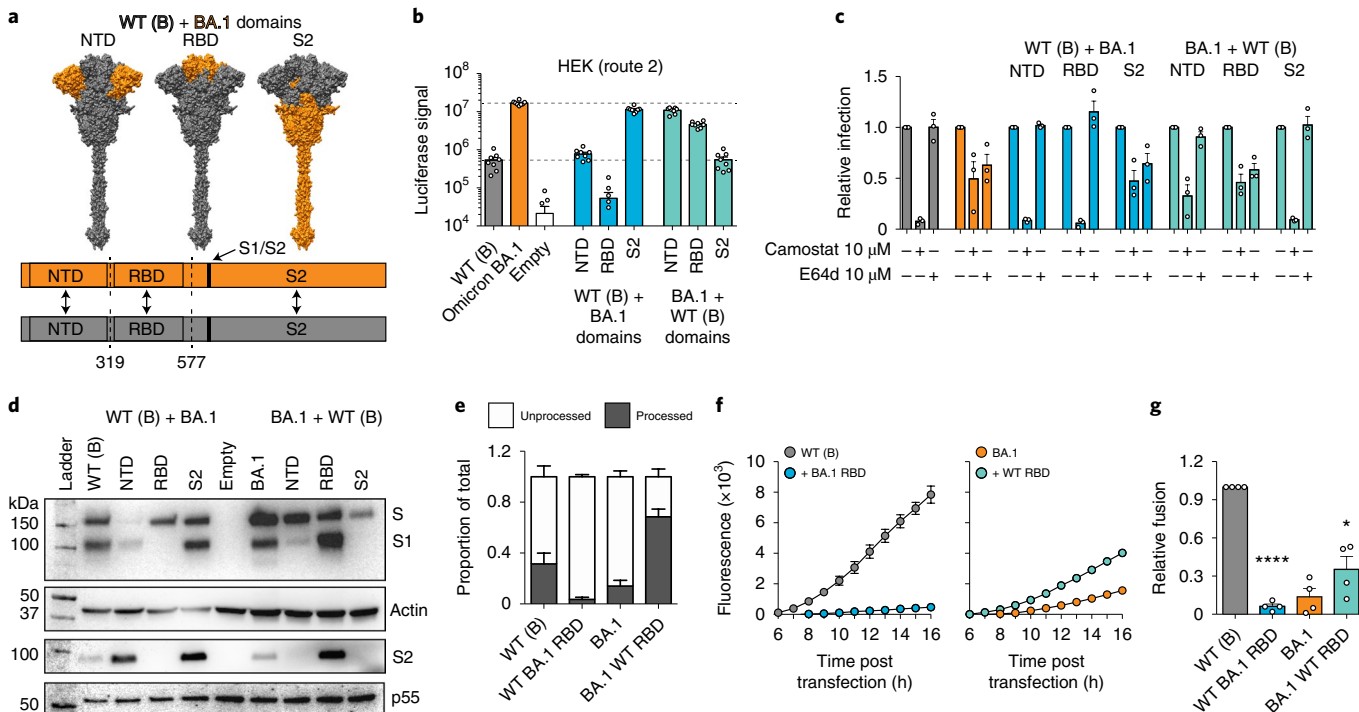

**Fig. 6 | Determinants of Omicron spike biology. a**, Top: schematic representation of domain swap constructs between WT (lineage B) and Omicron spike. Bottom: linear representation illustrates the junction points between the chimeric proteins. **b**, Domain swap pseudotype infection of HEK cells. Data represent mean luciferase values from a representative experiment ($n = 8$ technical repeats, data are representative of 3 independent experiments). **c**, Sensitivity of domain swap pseudotypes to camostat and E64d in A549-ACE2-TMPRSS2 cells. Data are expressed relative to the untreated control for each virus. Values represent the mean of 3 biological repeats. **d**, Immunoblot analyses of lysates from HEK cells producing domain swap pseudotypes. Blots were probed for spike (S1 and processed S2), actin loading control and HIV Gag p55 transfection control. The actin control relates to the S1 blot, while the p55 control relates to the processed S2 blot. **e**, Quantification of spike proteolytic processing in WT (lineage B), Omicron BA.1 and the reciprocal RBD swaps. Data are expressed relative to the total for a given spike and represent the mean of 5 independent blots. **f**, Cell fusion assay using cells transfected with WT (lineage B) (left), Omicron BA.1 (right) and the reciprocal RBD swap spikes. Plot displays mean GFP fluorescence signal from 1 representative experiment. **g**, Cell–cell fusion (as in **f**). Data are expressed relative to fusion by WT (lineage B) spike (at 16 h) and are the mean of 4 independent experiments. One-way ANOVA, *$P < 0.01$, ****$P < 0.0001$. In all plots, error bars represent s.e.m.

within and juxtaposed to the S1/S2 cleavage boundary (H655Y, N679K and P681H). Pseudotypes bearing the domain swap spikes were used to characterize entry pathway with the tools outlined in Fig. 5. Experiments in HEK cells suggest that the S2 portion of spike determines efficient entry via the endosomal route, as evidenced by increased and decreased infection by the respective S2 swaps (Fig. 6b). Notably, in this setting Omicron BA.1 RBD, presented in the ancestral spike background, was deleterious, suggesting a necessity for compensating mutations elsewhere in spike. Infection of A549-ACE2-TMPRSS2 confirmed that S2 determined endosomal entry, as evidenced by alterations in sensitivity to Camostat and E64d (Fig. 6c). In this experiment, the ancestral NTD, presented in the Omicron BA.1 background, gave an intermediate phenotype, suggesting that this region may also contribute to efficient endosomal entry.

Spike immunoblots were also performed using antibodies targeting both S1 and S2 to evaluate proteolytic processing. Surprisingly, these analyses indicated that the RBD harboured the master determinant of reduced proteolysis in Omicron BA.1 spike (Fig. 6d). This was clearest when comparing the reciprocal RBD swaps; Omicron BA.1 RBD prevented processing of ancestral spike, whereas ancestral WT (lineage B) RBD enabled highly efficient processing of Omicron BA.1 spike (Fig. 6e). Importantly, there was no correlation between efficient entry via the endosome and spike proteolysis. For example, Omicron spike bearing the ancestral RBD exhibited preferential entry via the endosome but had a highly processed spike.

As previous experiments with trypsin (Fig. 5m,n) suggested that the deficiency in syncytia formation associated with the Omicron spike was caused by reduced proteolysis, we reasoned that RBD swapping may modulate cell–cell fusion. Accordingly, the ancestral spike with the Omicron RBD was unable to mediate cell–cell fusion, whereas the reciprocal swap rescued fusion by the Omicron spike, albeit not to the same level as WT (lineage B) spike (Fig. 6f,g).

These experiments suggest that the phenotype attributed to the Omicron spike is determined by an interplay between mutations across multiple domains, possibly underpinned by epistasis and allostery. However, the major facets of Omicron spike biology can be mapped to distinct regions, with the S2 portion mediating efficient endosomal entry, while the RBD confers reduced proteolytic processing and associated defects in syncytia formation.

## Discussion

The Omicron variant represents a major change in biological function and antigenicity of SARS-CoV-2. In this study, we demonstrate substantial immune escape of the Omicron variant. We present clear evidence of vaccine failure in dual-vaccinated individuals and partial restoration of immunity following a third booster dose of mRNA vaccine. In addition, we demonstrate a shift in the SARS-CoV-2 entry pathway from cell surface fusion triggered by TMPRSS2 to cathepsin-dependent fusion within the endosome. Subsequent to the submission of this study for publication, aspects of these findings have been confirmed by other groups[55–67]. This fundamental

biological shift may affect the pathogenesis and severity of disease and requires further evaluation in population-based studies.

Using sera from double-vaccine recipients, Omicron BA.1 and BA.2 variants were found to be associated with a drop in neutralization greater in magnitude than that reported in all other variants of concern (including Beta and Delta). Boosting enhanced neutralizing responses to both the vaccine strain (WT (B)) and Omicron, particularly in recipients of ChAdOx1, but did not completely overcome the inherent immune escape properties of Omicron. In contrast, T-cell immunity in vaccine recipients measured by IFN-γ ELISpot stimulated by either WT (lineage B) or Omicron spike-derived peptides showed no significant difference, in agreement with the prediction that only 14% of CD8+ and 28% of CD4+ T-cell epitopes are likely to be affected by key Omicron mutations[12].

Vaccine effectiveness population data reflected the drop in immunity suggested by the neutralization experiments; the probability of infection with Omicron versus infection with the preceding Delta variant was significantly higher in double-vaccine recipients, in agreement with the neutralization data. A third dose of mRNA vaccine substantially reduced the probability of infection but did not fully restore immunity. Importantly, we did not assess the impact of vaccination on clinical severity of disease, which is likely to be much higher than detection of infection. Protection against severe disease is longer lasting than prevention of infection.

The emergence of a highly transmissible variant that is associated with escape from vaccine-induced immune responses means that over time, variant-specific vaccines may be required if associated disease severity is high, either directed at the general population or vulnerable groups. Early indications in young people are that Omicron infection is 40–70% less severe than Delta infection[68,69]. Similar calculations in the most vulnerable part of the population over the age of 40 years are awaited.

Genotypic changes in new variants have previously been shown to alter viral phenotype by modulating innate immune responses as well as evasion of the adaptive immune response[70,71]. Additionally, mutations can alter spike functionality to impact transmission and pathogenesis[24]. Such changes may have provided emergent viruses with a selective advantage in lung cells and primary human airway epithelial cells. Enhanced spike activation by the plasma membrane protease TMPRSS2 may have enabled more rapid cell surface fusion[44]. In this study, we found that the Omicron variant had switched entry pathway preference to use TMPRSS2-independent endosomal fusion—a major change in the biological behaviour of the virus and probably enabling alterations in tissue tropism. This switching of entry pathway was accompanied by alterations in proteolytic processing and reduced syncytia formation in infected cells, probably to limit cell-to-cell transmission and pathogenesis. These features of Omicron map to different regions of spike, with S2 determining endosomal entry and the RBD controlling proteolysis and syncytia formation. This is surprising as there was a previous assumption that these characteristics are directly linked; these data demonstrate a complex relationship between domains and suggest that there is still much to learn about the mechanics of spike function and the relationship of such changes to clinical disease.

In conclusion, it is important to note that even a variant that is less virulent with a very high transmission rate may continue to present a risk to older people and those with co-morbidities, especially those with immunosuppression or who are unvaccinated. Moreover, our work demonstrates that SARS-CoV-2 exhibits high antigenic and functional plasticity; further fundamental shifts in transmission and disease should be anticipated.

## Methods

**Ethics statement.** All participants in the DOVE study gave written informed consent to take part in the study which was approved by the North-West Liverpool Central Research Ethics Committee (REC reference 21/NW/0073). Residual nasopharyngeal swabs of patients infected with Omicron were collected with biorepository ethical approval (NHS Lothian reference 20/ES/0061). Derogated ethical approval for the use of demographic data for the EVADE study was granted by the NHS GG&C SafeHaven committee (GSH/21/IM/001).

**Cells.** Calu-3 cells (ATCC HTB-55) are human lung adenocarcinoma epithelial cells. Caco-2 cells (CVR cytology cell bank) are from an immortalized cell line derived from human colorectal adenocarcinoma, primarily used as a model of the intestinal epithelial barrier. A549 cells (ATCC CCL-185) are from a human alveolar adenocarcinoma line and were a generous gift from Prof. Ben Hale, validated by short tandem repeat analysis (Eurofins). A549 cells were modified to stably express human ACE2 and TMPRSS2. HEK293T cells were used in pseudotype production. African green monkey kidney cells (Vero) were used to propagate the reverse genetics-derived viruses. Baby Hamster Kidney clone 21 cells (BHK-21 ATCC CCL-10) and Vero ACE2 TMPRSS2[72] cells were used in the isolation of live Omicron SARS-CoV-2. All cell lines were maintained at 37 °C and 5% CO2 in DMEM supplemented with 10% foetal bovine serum (FBS), except for Calu-3 cells which were supplemented with 20% FBS. Human reconstituted upper airway epithelium cells (Mucilair, abbreviated hNECs in this Article) were purchased from Epithelix and maintained in Mucilair complete culture basal medium (Epithelix) at an air–liquid interface.

**Generation of BHK-21 cell line expressing human ACE2 receptor.** Lentiviral vectors encoding human *ACE2* (GenBank NM_001371415.1) were produced as described previously[72]. BHK-21 cells were transduced with the ACE2-encoding lentivirus and selected in medium containing 200 µg ml⁻¹ of hygromycin B. A pool of hygromycin-resistant cells, BHK-ACE2, was used in this study.

**Generation of cell lines used for fusion assays.** Retrovirus vectors were produced by transfecting HEK293T cells with plasmid pQCXIP-GFP1-10 (Addgene 68715) or pQCXIP-BSR-GFP11 (Addgene 68716)[42], alongside packaging vectors expressing murine leukemia virus gal-pol and vesicular stomatitis virus G protein using Lipofectamine 3000 (Invitrogen) according to the manufacturer's instructions. Cell supernatants were collected 24–48 h post transfection, pooled, clarified by centrifugation and filtered. One ml of each supernatant was used to transduce A549-Ace2-TMPRSS2 (AAT) cells[72] in the presence of Polybrene (Merck). At 2 d post transduction, the supernatant was replaced with selection medium (DMEM, 10% FBS, 1 µg ml⁻¹ puromycin) and cells incubated until complete death of the non-transduced control cells were observed. The resulting puromycin-resistant cells (termed AAT-GFP1-10 and AAT-BSR-GFP11) were used in fusion assays.

**Virus isolation from clinical samples.** Residual nasopharyngeal swabs of patients infected with Omicron were collected with biorepository ethical approval (NHS Lothian reference 20/ES/0061) in virus transport medium and resuspended in serum-free DMEM supplemented with 10 µg ml⁻¹ gentamicin, 100 units ml⁻¹ penicillin-streptomycin and 2.5 µg ml⁻¹ amphotericin B to a final volume of 1.5 ml. Virus isolation was attempted in BHK-21 cells stably expressing the human ACE2 protein (BHK-hACE2) and VERO cells stably expressing ACE2 and TMPRSS2 (VAT)[72]. The infected cells were incubated at 37 °C and monitored for signs of CPE and the presence of viral progeny in the medium by quantitative PCR with reverse transcription (RT-qPCR). While no CPE was observed in any of the infected cells, RT-qPCR at 5 d post infection confirmed the presence of the virus derived from two of the five samples (referred to hereafter as 204 and 205) in the medium of BHK-hACE2, but not VERO ACE2 TMPRSS2 cells (Extended Data Fig. 10a). An aliquot of the clarified medium containing approximately 4 × 10⁴ viral genomes of the P0 stocks of samples 204 and 205 was used to infect VAT, BHK-ACE2 and Calu-3 cells. No CPE was observed in the infected cells but once again, virus replication was confirmed in BHK-hACE2 and Calu-3 by RT-qPCR. Supernatants from infected Calu-3 cells (termed P1) at 3 dpi were collected and virus titrated by both focus-forming assay and RT-qPCR. The virus reached more than 100-fold higher titres in Calu-3 cells compared with BHK-hACE2 (Extended Data Fig. 10b). Further passage of sample 205-derived P1 virus in both Calu-3 and Caco-2 yielded equivalent genome copy numbers in both cell lines (Extended Data Fig. 10b). CPE was observed at 3 dpi in both Calu-3 and Caco-2 cells (not shown). The medium of infected Calu-3 and Caco-2 cells (termed P2) was collected at 4 dpi, titrated and used in subsequent experiments.

Virus samples were sequenced essentially as previously described[73]. RNA was extracted from 250 µl of cell culture supernatant using TRIzol LS (Thermo Fisher) and purified with RNAeasy mini kit (Qiagen). RNA (11 µl) was reverse transcribed using Superscript III (Invitrogen) with random hexamers. Following second strand synthesis with NEBNext Ultra II Non-Directional RNA Second Strand (New England BioLabs), libraries were prepared using the KAPA library prep kit (KAPA Biosystems) with index tagging using KAPA HiFi HotStart polymerase and unique dual indices (New England Biolabs, set 3). The resulting libraries were quantified by Qubit (Thermo Fisher) and TapeStation (Agilent) and pooled at equimolar concentrations for sequencing on the Illumina NextSeq550 platform, using a mid-output 300-cycles cartridge. Of the reads, 85.8% had a Q score above 30. Illumina paired-end reads were aligned to the Wuhan-Hu-1 reference genome (MN908947.3) using bwa[74], followed by consensus calling with iVar[75]. Sample 205 yielded a complete genome sequence, which was confirmed to be Omicron (lineage BA.1) by Pango[76] (GISAID id: EPI_ISL_10666879).

**Generation of recombinant SARS-CoV-2 using reverse genetics.** Recombinant viruses described in this study were generated using transformation-associated recombination in yeast as we described previously[77]. SARS-CoV-2 recombinant viruses carrying the Delta or Omicron variant spike within the ancestral WT lineage B backbone were assembled by transformation-associated recombination in yeast using a set of relevant overlapping complementary DNA fragments to assemble the modified genomes. RNA transcribed in vitro from the recombinant genomes was used to rescue the viruses following transfection into BHK cells stably expressing ACE2 and SARS-CoV-2 N protein. Two clones of each rescued virus were passaged (P1) into VERO E6 cells and their genomes verified by sequencing using Oxford Nanopore as described above[73].

**Measurement of SARS-CoV-2, HcoVs and influenza antibody response by electrochemiluminescence.** IgG antibody titres were measured quantitatively against SARS-CoV-2 trimeric spike (S) protein, N-terminal domain (NTD), receptor-binding domain (RBD) or nucleocapsid (N); human seasonal coronaviruses (HcoVs) 229E, OC43, NL63 and HKU1; and influenza A (Michigan H1, Hong Kong H3 and Shanghai H7) and B (Phuket HA and Brisbane) using MSD V-PLEX COVID-19 Coronavirus Panel 2 (K15369) and Respiratory Panel 1 (K15365) kits. MSD-ECL assays were performed according to manufacturer instructions. Briefly, 96-well plates were blocked for 1 h. Plates were then washed, samples were diluted 1:5,000 in diluent and added to the plates along with serially diluted reference standard (calibrator) and serology controls 1.1, 1.2 and 1.3. After incubation, plates were washed and SULFO-TAG detection antibody added. Plates were washed and immediately read using a MESO Sector S 600 plate reader. Data were generated by Methodological Mind software and analysed using MSD Discovery Workbench (v4.0). Results are expressed as MSD arbitrary units per ml (a.u. ml$^{-1}$). Reference plasma samples yielded the following values: negative pool – spike 56.6 a.u. ml$^{-1}$, NTD 119.4 a.u. ml$^{-1}$, RBD 110.5 a.u. ml$^{-1}$ and nucleocapsid 20.7 a.u. ml$^{-1}$; SARS-CoV-2 positive pool – spike 1,331.1 a.u. ml$^{-1}$, NTD 1,545.2 a.u. ml$^{-1}$, RBD 1,156.4 a.u. ml$^{-1}$ and nucleocapsid 1,549.0 a.u. ml$^{-1}$; NIBSC 20/130 reference – spike 547.7 a.u. ml$^{-1}$, NTD 538.8 a.u. ml$^{-1}$, RBD 536.9 a.u. ml$^{-1}$ and nucleocapsid 1,840.2 a.u. ml$^{-1}$.

**Measurement of virus neutralizing antibodies using viral pseudotypes.** Pseudotype-based neutralization assays were carried out as described previously[78–80]. Briefly, HEK293, HEK293T and 293-ACE2[79] cells were maintained in DMEM supplemented with 10% FBS, 200 mM L-glutamine, 100 μg ml$^{-1}$ streptomycin and 100 IU ml$^{-1}$ penicillin. HEK293T cells were transfected with the appropriate SARS-CoV-2 S gene expression vector (wild type or other variant) in conjunction with p8.91[81] and pCSFLW[82] using polyethylenimine (PEI, Polysciences). HIV (SARS-CoV-2) pseudotypes containing supernatants were collected 48 h post transfection, aliquoted and frozen at −80 °C before use. S gene constructs bearing the WT (lineage B, corresponding to the Wuhan-Hu-1 strain), B.1 lineage (D614G) and Omicron (BA.1 and BA.2) S genes were based on the codon-optimized spike sequence of SARS-CoV-2 and generated by GeneArt (Thermo Fisher). Constructs bore the following mutations relative to the WT lineage B sequence (Wuhan-Hu-1, GenBank: MN908947): B.1 – D614G; Omicron (BA.1, B.1.1.529) – A67V, Δ69–70, T95I, G142D/Δ143–145, Δ211/L212I, ins214EPE, G339D, S371L, S373P, S375F, K417N, N440K, G446S, S477N, T478K, E484A, Q493R, Q496S, Q498R, N501Y, Y505H, T547K, D614G, H655Y, N679K, P681H, N764K, D796Y, N856K, Q954H, N969K and L981F; Omicron (BA.2) – T19I, Δ24–26/A27S, G142D, V213G, G339D, S371F, S373P, S375F, T376A, D405N, R408S, K417N, N440K, S477N, T478K, E484A, Q493R, Q498R, N501Y, Y505H, D614G, H655Y, N679K, P681H, N764K, D796Y, Q954H and N969K. BA.1.1 was prepared by site-directed mutagenesis (Q5 site-directed mutagenesis kit, New England Biolabs) of the BA.1 construct to introduce R346K. 293-ACE2 target cells were maintained in complete DMEM supplemented with 2 μg ml$^{-1}$ puromycin.

Neutralizing activity in each sample was measured by a serial dilution approach. Each sample was serially diluted in triplicate from 1:50 to 1:36,450 in complete DMEM before incubation with HIV (SARS-CoV-2) pseudotypes, incubated for 1 h and plated onto 239-ACE2 target cells. After 48–72 h, luciferase activity was quantified by the addition of Steadylite Plus chemiluminescence substrate and analysis on a Perkin Elmer EnSight multimode plate reader. Antibody titre was then estimated by interpolating the point at which infectivity had been reduced to 50% of the value for the no serum control samples.

**ELISpot assays.** SARS-CoV-2 peptide pools were designed and provided by members of the PITCH consortium as previously described[33]. Pools of spike protein (S1 and S2) from Wuhan and Omicron (BA.1) strains were used in this study.

Fresh peripheral blood mononuclear cells (PBMCs) from all study subjects were isolated by density gradient centrifugation over Histopaque-1077 (p = 1.077 g ml$^{-1}$; Sigma). Plasma was collected and stored in aliquots at −80 °C. Buffy coat containing the PBMCs was collected and washed twice with PBS. Cells were either processed fresh, or frozen and stored in liquid nitrogen before use. PBMCs were seeded in recombinant anti-IFN-γ-coated PVDF 96-well plates (MabTech) at 200,000 cells per well in 50 μl R10 medium (Glutamax + RPMI (Gibco), 10% heat-inactivated foetal calf serum (Gibco), 10 mM HEPES buffer

(Gibco) and antibiotics (100 U ml$^{-1}$ penicillin-streptomycin)). Peptides were diluted in 50 μl R10 medium and added to the wells at a final concentration of 2 μg ml$^{-1}$. As a negative and a positive control, cells were stimulated with an equivalent volume of DMSO and 1:1,000 anti-CD3 (mAb CD3-2, Mabtech), respectively. The plates were incubated for 18 h at 37 °C and 5% $CO_2$. The plates were then washed with PBS and incubated with 1 μg ml$^{-1}$ biotin-labelled detection antibody (7-B6-1, Mabtech) for 2 h, followed by incubation with 1:1,000 dilution streptavidin-alkaline phosphatase (Mabtech) and subsequent incubation with BCIP/NBT-plus substrate solution (Mabtech). After spot development, the plates were washed extensively with tap water. The plates were dried and spots were quantified using a VIRUSpot reader (AID). IFN-γ-producing cells were expressed as spot-forming cells per million (SFC per million), where the reading from each well was subtracted with the median SFC per million of the DMSO-stimulated wells. Samples with a DMSO control reading of ≤50 SFC per million were excluded.

**Domain swap constructs.** We took advantage of fortuitous restriction sites (Bsu36I, PflMI and EcoNI) that are common to the ancestral WT (lineage B) and Omicron spike plasmids to perform reciprocal domain swaps. The Omicron mutations found within each swap are as follows. *NTD*: A67V, Δ69–70, T95I, G142D/Δ143–145, Δ211/L212I and ins214EPE; *RBD*: G339D, S371L, S373P, S375F, K417N, N440K, G446S, S477N, T478K, E484A, Q493R, G496S, Q498R, N501Y, Y505H and T547K; *S2*: D614G, H655Y, N679K, P681H, N764K, D796Y, N856K, Q954H, N969K and L981F. Sanger sequencing was used to confirm each construct.

**Protease inhibitor studies.** To selectively inhibit either cell surface or endosomal fusion of SARS-CoV-2, cells were pre-treated for 1 h with 10 μM of either Camostat mesylate (Camostat) or E64d before inoculation with pseudotyped virus or infection with $4 \times 10^5$ Orf1a genome copies per well of indicated SARS-CoV-2 VOCs. In the pseudotype studies, spike proteins from Alpha and Delta VOCs, and Guangdong isolate Pangolin coronavirus (GISAID ref EPI_ISL_410721) were used as controls.

**Viral RNA extraction and RT-qPCR.** Viral RNA was extracted from culture supernatants using the RNAdvance blood kit (Beckman Coulter Life Sciences) following the manufacturer's recommendations. RNA was used as template to detect and quantify viral genomes by duplex RT-qPCR using a Luna Universal Probe one-step RT-qPCR kit (New England Biolabs, E3006E). SARS-CoV-2-specific RNAs were detected by targeting the N1 gene from the Centers for Disease Control and Prevention panel as part of the SARS-CoV-2 Research Use Only qPCR Probe kit (Integrated DNA Technologies) and the ORF1ab gene using the following set of primers and probes: SARS-CoV-2_Orf1ab_Forward 5′ GACATAGAAGTTACTGG&CGATAG 3′, SARS-CoV-2_Orf1ab_Reverse 5′ TTAATATGACGCGCACTACAG 3′, SARS-CoV-2_Orf1ab_Probe ACCCCGTGACCTTGGTGCTTGT with HEX/ZEN/3IABkFQ modifications. SARS-CoV-2 RNA was used to generate a standard curve, and viral genomes were quantified and expressed as number of Orf1ab RNA molecules per ml of supernatant. All runs were performed on the ABI7500 Fast instrument and results analysed with the 7500 Software v2.3 (Applied Biosystems, Life Technologies).

**Genome sequencing.** Sequencing was carried out by the UK public health agencies (UKHSA/PHE, PHS, PHW and PHNI) and by members of the COG-UK consortium using the ARTIC protocol as previously described.

**Replication curve.** Calu-3 cells were seeded in a 96-well plate at a cell density of $3.5 \times 10^4$ cells per well. Cells were infected with the indicated viruses using the equivalent of $2 \times 10^4$ Orf1ab genome copies per well in serum-free RPMI-1640 medium (Gibco). After 1 h of incubation at 37 °C, cells were washed three times and left in 20% FBS RPMI-1640 medium. Supernatants were collected at different times post infection and viral RNA extracted and quantified as described above. Before infection, hNECs were washed with serum-free DMEM (SF-DMEM) to remove excess mucus and debris. Cells were infected with $1 \times 10^5$ Orf1ab genome copies per well in SF-DMEM and incubated for 1 h at 37 °C and 5% $CO_2$. The inoculum was then removed and the cells washed once with SF-DMEM before incubation at 37 °C and 5% $CO_2$. Samples were collected at the indicated time points by adding 100 μl SF-DMEM and incubating for 20 min at 37 °C before collection in 150 μl lysis buffer from the RNAdvance blood kit (Beckman Coulter Life Sciences) for viral RNA extraction.

**Fusion assay.** AAT-GFP1-10 and AAT-BSR-GFP11 cells were trypsinized and mixed at a ratio of 1:1 to seed a total of $2 \times 10^4$ cells per well in black 96-well plate (Greiner) in FluoroBrite DMEM medium (Thermo Fisher) supplemented with 2% FBS. Next day, cells were infected with the indicated viruses using the equivalent of $10^6$ Orf1a genome copies per well in FluoroBrite DMEM, 2% FBS or transfected with 0.1 μg DNA per well of spike plasmid using Lipofectamine LTX (Thermo Fisher). The GFP signal was acquired for the following 20 h using a CLARIOStar Plus (BMG LABTECH) equipped with an atmospheric control unit to maintain 37 °C and 5% $CO_2$. Data were analysed using MARS software and plotted with GraphPad Prism 9 software. At 22 h post infection, cells were fixed in 8% formaldehyde, permeabilized with 0.1 % Triton X-100 and stained with sheep

anti-SARS-CoV-2 N (1:500) antiserum[72], followed by Alexa Fluor 594 donkey anti-sheep IgG (H+L) (1:500, Invitrogen) and DAPI (1:4,000, Sigma). Cell images were acquired using EVOS Cell Imaging Systems (Thermo Fisher). For the trypsin experiments, cells were prepared for the fusion assay and transfected with the spike expression plasmid. At 8 h post transfection, the medium was replaced with serum-free FluoroBrite DMEM supplemented with Trypsin-Ultra (New England Biology) at the indicated concentration. The GFP signal was acquired for the following 20 h as described above.

**Immunoblotting.** Cells were lysed in ice-cold lysis buffer containing 50 mM Tris/HCl pH 7.5, 1 mM EGTA, 1 mM EDTA, 1% (v/v) Triton X-100, 1 mM sodium ortho-vanadate, 50 mM NaF, 5 mM sodium pyrophosphate, 0.27 M sucrose, 10 mM sodium 2-glycerophosphate, 1 mM phenylmethylsulphonyl fluoride, 1 mM benzamidine[83] and 1X NuPAGE LDS sample buffer (NP0007, Thermo Fisher). Samples were then subjected to SDS–PAGE and immunoblotted for ACE2 (Abcam, ab108252), human Cathepsin-L (AF952, R&D Systems), TMPRSS2 (14437-1-AP, Proteintech), SARS-CoV-2 spike protein S1-NTD (E7M5X, Cell Signalling Technology), mouse anti-SARS-CoV-2 spike protein S2 (clone 1A9, GeneTex), sheep anti-SARS-CoV-2 nucleocapsid protein 3rd bleed[72], GAPDH (2118, Cell Signalling Technology), mouse anti-beta actin (AC-15, Abcam) or mouse anti-p55 (EVA365, NIBSC). Anti-rabbit IgG (H + L) DyLight 800 conjugate, anti-mouse IgG (H+L) DyLight 680 conjugate, donkey anti-goat IgG DyLight 800 (Thermo Fisher, SA5-10090), rabbit anti-sheep IgG451 (H+L) DyLight 800 (Thermo Fisher, SA5-10060) or anti-mouse HRP secondary antibodies were used for immunoblotting before protein visualization using the Odyssey CLx imager (Li-Cor) or ChemiDoc XRS (BioRad).

**Demographic data.** Data for the EVADE study were available using the NHS GG&C SafeHaven platform and included vaccination status (dates and product names for each dose), demographic data (age, sex and SIMD quartile), comorbidity (shielding and immunosuppression status) and dates of positive and negative PCR tests for 1.2 million inhabitants over 18 years of age in the NHS GG&C area, from 1 March 2020 up to 9 February 2022. Data were matched by Community Health Index number and pseudonymized before analysis. Derogated ethical approval was granted by the NHS GG&C SafeHaven committee (GSH/21/IM/001).

**Vaccine effectiveness.** A logistic additive regression model was used to estimate relative vaccine effectiveness against the Omicron variant as it emerged in a population of 1.2 million people in NHS GG&C, the largest health board in Scotland. Infection status for Omicron and Delta was modelled by number and product type of vaccine doses, previous infection status, sex, SIMD quartile and age on 31st October 2021.

Omicron infections were identified using three data streams: confirmed SGTF, allele-specific PCR and Pango lineage assignments from sequencing data. SGTF samples with Delta lineage assignments were assigned as Delta infections and samples with S gene presence were also assigned as Delta infections since this was appropriate during our study period. We used a subset of data points comprising those tested between 6 December 2021 and 26 December 2021, and confirmed either positive with Delta, positive with Omicron, or negative.

A small number of individuals who received ChAdOx1 as a third dose were removed on the assumption that the majority were part of the COV-BOOST clinical trial, the results of which are published elsewhere. We also removed anyone who received a vaccine other than ChAdOx1, BNT162b2 or mRNA-1273 or whose brand was unknown due to data entry error. We removed individuals who tested positive in the 90 d before the study period, since it would not be possible for those individuals to have a new infection recorded. To avoid introducing bias by excluding those who changed vaccine status during the study period and who were less likely to be infected than those who did not change status (due to dosing eligibility criteria), we fixed vaccine status as that at 14 d before the start of the study period. Because those who were confirmed infected were not eligible for additional doses until 28 d after their positive test, those who received additional doses were less infected on average than those who did not. Their exclusion increased the infection rate among cohorts, especially double-dosed cohorts who were most likely to seek additional doses during the study period.

**Serum samples.** Serum and PBMC samples were collected from healthy participants in the COVID-19 Deployed Vaccine Cohort Study (DOVE), a cross-sectional post-licensing cohort study to determine the immunogenicity of deployed COVID-19 vaccines against evolving SARS-CoV-2 variants. Adult volunteers (308) aged at least 18 years were recruited to the study 14 d or more after a second or third dose of vaccine. All participants gave written informed consent to take part in the study. The DOVE study was approved by the North-West Liverpool Central Research Ethics Committee (REC reference 21/NW/0073).

**Structural modelling.** The file 6vsb_1_1_1.pdb containing a complete model of the full-length glycosylated spike homotrimer in open conformation with one monomer having the receptor-binding domain in the 'up' position was obtained from the CHARMM-GUI Archive[84,85]. This model is itself generated on the basis of a partial spike cryo-EM structure (PDB ID: 6VSB). For visualization,

the model was trimmed to the ectodomain (residues 14–1,164), and the signal peptide (residues 1–13) and glycans were removed using this structural model and the closed conformation equivalent (6vxx_1_1_1.pdb). Residues belonging to the receptor-binding site were identified as those with an atom within 4 Å of an ACE2 atom in the bound RBD-ACE2 structure (PDB ID: 6M0J[86]), and Alpha carbon-to-Alpha carbon distances between these residues in the 'up' RBD and all other spike residues were calculated. Antibody accessibility scores for open and closed conformations were calculated using BEpro[87]. Figures were prepared using PyMol[88].

**Epidemiological description of the emergence of the Omicron variant in the UK.** On 27 November 2021, the UK Health Security Agency detected 2 cases of Omicron in England, the following day 6 Scottish cases were detected by community (Pillar 2) sequencing. Over the next 10 d (to 8 December 2021), a further 95 genome sequences were obtained. These sequences were aligned by mapping to the SARS-CoV-2 reference Wuhan-Hu-1 using Minimap2[89]. Before phylogenetic analysis, 85 sites exhibiting high genetic variability due to data quality issues in overseas sequencing labs were excluded using a masking script in Phylopipe (https://github.com/cov-ert/phylopipe). A phylogenetic tree was constructed with the maximum likelihood method FastTree2[90] using a JC+CAT nucleotide substitution model.

Due to the rapid spread of Omicron and low genetic diversity, the genome sequences are highly related, with mean genetic divergence of 1 single nucleotide polymorphism (SNP) and maximum of 7 SNPs.

The phylogenetic relationship to Omicron sequences from other countries (not shown) is consistent with multiple introductions associated with travel to South Africa, followed by community transmissions within Scotland. Among the Scottish samples diverged from the tree backbone, there were a number identified that are genetically divergent, that is, >2 SNPs from the nearest Scottish sample. Moreover, comparison to the wider international collection of Omicron samples revealed that they were more closely related to genomes from other countries than other Scottish samples. These samples therefore probably represent independent introductions to Scotland, but without more detailed epidemiological data, the number of introductions is unknown. Where there are indistinguishable samples in the phylogeny from Scotland and elsewhere in world, importation cannot be ruled out as a source of these samples in Scotland, rather than transmission from an established population circulating in Scotland.

Within Scotland, cases are spread across 9 separate Health Boards and distributed throughout the phylogeny. Basal Scottish genomes were sampled in 7 different Health Boards, most of them from NHS GG&C (47%) and NHS Lanarkshire (25%). Notably, among these earliest samples are cases that were epidemiologically linked to early spreading events. All but one of these samples were found on this basal branch and are indistinguishable, which is consistent with transmission at these events.

**Reporting summary.** Further information on research design is available in the Nature Research Reporting Summary linked to this article.

## Data availability
Datasets/experimental data generated and/or analysed during the current study are appended as Supplementary Information. However, restrictions apply to the availability of the clinical data, which were used under ethical approvals for the current study, and so are not publicly available. Anonymized data used for estimating vaccine effectiveness are available with permission of the NHS Greater Glasgow & Clyde SafeHaven. Clinical samples are restricted for use under the ethical approvals obtained for their use. Biological materials including cell lines are available on reasonable request from the corresponding authors. Source data are provided with this paper.

## Code availability
Codes used in this analysis are available in the study's GitHub repository: https://github.com/centre-for-virus-research/Omicron.

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

## Acknowledgements

We thank the participants of the DOVE study and T. McSorley and her nursing team at the NHS GG&C clinical research facility; A. Hamilton, L. Stirling and C. Mayor from the NHS GG&C SafeHaven team for their invaluable input in facilitating this study; P. Olmo for administrative support, and C. Robertson and A. Sheikh for statistical advice; and all the researchers who have shared genome data openly via the Global Initiative on Sharing All Influenza Data (GISAID). Funding was provided by Health Data Research UK (HDR UK) for the Evaluation of Variants Affecting Deployed COVID-19 Vaccine (EVADE) study (E.C.T., S.R., O.A.M., C.W. and B.J.W.; grant code: 2021.0155). This research is part of the Data and Connectivity National Core Study, led by Health Data Research UK in partnership with the Office for National Statistics and funded by UK Research and Innovation (grant ref MC_PC_20058). This work was also supported by The Alan Turing Institute via 'Towards Turing 2.0' EPSRC Grant Funding. COG-UK is supported by funding from the Medical Research Council (MRC, part of UK Research & Innovation (UKRI)), the National Institute of Health Research (NIHR; grant code: MC_PC_19027) and Genome Research Limited, operating as the Wellcome Sanger Institute (R.M.P., D.L.R. and E.C.T.). Medical Research Council (MRC) provided funding for both the COVID-19 DeplOyed VaccinE (DOVE) study (grant code: MCUU1201412) and COG-UK (E.C.T.). A.d.S.F., J.H., R.O., J.G., E.C.T., N.L. and D.L.R. were funded by the Medical Research Council (MRC; grant code: MC_UU_12014/12). W.T.H. was supported by the MRC (grant codes MR/R024758/1 and MR/W005611/1). The G2P-UK National Virology Consortium was funded by UK Research and Innovation (UKRI) award MR/W005611/1 (M.P., E.C.T., A.H.P. and D.L.R.). D.L.R. was funded by Wellcome Trust (grant code: 220977/Z/20/Z). N.L. and B.J.W. were funded by the Biotechnology and Biological Sciences Research Council (grant codes: BBSRC, BB/R004250/1 and BB/R019843/1). J.G. was funded by a Wellcome Trust and Royal Society Sir Henry Dale Fellowship (grant code: 107653/Z/15/A). D.J.P. was funded by UKRI through the JUNIPER consortium (grant number MR/V038613/1). The PITCH Consortium is funded by the United Kingdom Department of Health and Social Care.

## Author contributions

E.C.T., J.G., M.P., D.L.R. and B.J.W. conceptualized the project; B.J.W., J.G., O.A.M., C.W., N.L., G.D.L., W.F., S. Scott, M.M., A.S., W.T.H., C.D., R.O., J.H., D.J.P., K.P., A.d.S.F., G.Y., S. Shaaban, R.M.P., P.R.M., A.H.P., J.H., D.L.R., P.K., S.D., PITCH Consortium, COG-UK Consortium and SR developed the methodology; B.J.W., J.G., O.A.M., C.W., N.L., G.D.L., W.F., S. Scott, M.M., A.S., S.A., E.V., W.T.H., C.D., R.O., J.H., P.H., V.S., D.J.P., K.P., A.d.S.F., G.Y., S. Shaaban, M.T.G.H., R.G., K.T., A.H.P., J.H., D.L.R., M.P., D.C., V.C., P.A., K.S., L.T. and S.R. conducted the investigations; B.J.W., J.G., O.A.M., C.W., G.D.L., W.T.H. and D.L.R. conducted visualization; E.C.T., D.L.R., S.R., B.J.W. and J.G. acquired funding; E.C.T., D.L.R., S.R. and J.H. administered the project; B.J.W., D.L.R., M.P., S.R., J.H. and E.C.T. supervised the project; E.C.T., B.J.W., J.G. and G.Y. wrote the original draft; B.J.W., J.G., O.A.M., C.W., S.A., W.T.H., D.L.R., M.P. and E.C.T. reviewed and edited the manuscript.

## Competing interests

S.D. declares fees as a Scientific Advisor to the Scottish Parliament on COVID-19. The other authors declare no competing interests.

## Additional information

**Extended data** is available for this paper at https://doi.org/10.1038/s41564-022-01143-7.

**Correspondence and requests for materials** should be addressed to Brian J. Willett, Joe Grove or Emma C. Thomson.

## PITCH Consortium

**Susanna Dunachie[10,11,12,13], Paul Klenerman[10,11,14,15], Eleanor Barnes[10,11,14,15], Anthony Brown[10], Sandra Adele[10,12], Barbara Kronsteiner[10,12], Sam M. Murray[10], Priyanka Abraham[10], Alexandra Deeks[10], M. Azim Ansari[10], Thushan de Silva[16,17], Lance Turtle[18,19], Shona Moore[18], James Austin[18], Alex Richter[20,21], Christopher Duncan[22,23] and Rebecca Payne[22]**

[10]Peter Medawar Building for Pathogen Research, Nuffield Department of Clinical Medicine, University of Oxford, Oxford, UK. [11]Oxford University Hospitals NHS Foundation Trust, John Radcliffe Hospital, Oxford, UK. [12]Oxford Centre For Global Health Research, Nuffield Department of Clinical Medicine, University of Oxford, Oxford, UK. [13]Mahidol-Oxford Tropical Medicine Research Unit, Bangkok, Thailand. [14]Translational Gastroenterology Unit, University of Oxford, Oxford, UK. [15]NIHR Oxford Biomedical Research Centre, University of Oxford, Oxford, UK. [16]Department of Infection, Immunity and Cardiovascular Disease, University of Sheffield, Sheffield, UK. [17]Sheffield Teaching Hospitals NHS Foundation Trust, Sheffield, UK. [18]NIHR Health Protection Research Unit in Emerging and Zoonotic Infections, Institute of Infection, Veterinary and Ecological Sciences, University of Liverpool, Liverpool, UK. [19]Tropical and Infectious Disease Unit, Liverpool University Hospitals NHS Foundation Trust, Liverpool Health Partners, Liverpool, UK. [20]Institute of Cancer and Genomic Science, College of Medical and Dental Science, University of Birmingham, Birmingham, UK. [21]University Hospitals Birmingham NHS Foundation Trust, Birmingham, UK. [22]Translational and Clinical Research Institute Immunity and Inflammation Theme, Newcastle University, Newcastle, UK. [23]Department of Infection and Tropical Medicine, Newcastle upon Tyne Hospitals NHS Foundation Trust, Newcastle, UK.

## The COVID-19 Genomics UK (COG-UK) Consortium

**Amy Ash[24], Cherian Koshy[24], Beatrix Kele[25], Teresa Cutino-Moguel[25], Derek J. Fairley[26], James P. McKenna[26], Tanya Curran[26], Helen Adams[27], Christophe Fraser[28], David Bonsall[28], Helen Fryer[28], Katrina Lythgoe[28], Laura Thomson[28], Tanya Golubchik[28], Abigail Murray[29], Dawn Singleton[29], Shaun M. Beckwith[29], Anna Mantzouratou[30], Magdalena Barrow[30], Sarah L. Buchan[30], Nicola Reynolds[31], Ben Warne[32], Joshua Maksimovic[33], Karla Spellman[33], Kathryn McCluggage[33], Michaela John[33], Robert Beer[33], Safiah Afifi[33], Sian Morgan[33], Andrew Mack[34], Angela Marchbank[34], Anna Price[34], Arthur Morriss[34], Catherine Bresner[34], Christine Kitchen[34], Ian Merrick[34], Joel Southgate[34], Martyn Guest[34], Owen Jones[34], Robert Munn[34], Thomas R. Connor[34], Thomas Whalley[34], Trudy Workman[34], William Fuller[34], Amita Patel[35], Bindi Patel[35], Gaia Nebbia[35], Jonathan Edgeworth[35], Luke B. Snell[35], Rahul Batra[35], Themoula Charalampous[35], Angela H. Beckett[36], Ekaterina Shelest[36], Samuel C. Robson[36], Anthony P. Underwood[37], Ben E. W. Taylor[37], Corin A. Yeats[37], David M. Aanensen[37], Khalil Abudahab[37], Mirko Menegazzo[37], Amelia Joseph[38], Gemma Clark[38], Hannah C. Howson-Wells[38], Louise Berry[38], Manjinder Khakh[38], Michelle M. Lister[38], Tim Boswell[38], Vicki M. Fleming[38], Christopher W. Holmes[39], Claire L. McMurray[39], Jessica Shaw[39], Julian W. Tang[39], Karlie Fallon[39], Mina Odedra[39], Nicholas J. Willford[39], Paul W. Bird[39], Thomas Helmer[39], Lesley-Anne Williams[40], Nicola Sheriff[40], Sharon Campbell[40], Veena Raviprakash[40], Victoria Blakey[40], Christopher Moore[41], Fei Sang[41], Johnny Debebe[41], Matthew Carlile[41], Matthew W. Loose[41], Nadine Holmes[41], Victoria Wright[41], M. Estee Torok[42], William L. Hamilton[42], Alessandro M. Carabelli[43], Andrew Jermy[43], Beth Blane[43], Carol M. Churcher[43], Catherine Ludden[43], Dinesh Aggarwal[43], Elaine Westwick[43], Ellena Brooks[43], Georgina M. McManus[43], Katerina Galai[43], Ken Smith[43], Kim S. Smith[43], MacGregor Cox[43], Mireille Fragakis[43], Patrick Maxwell[43], Sarah Judges[43], Sharon J. Peacock[43], Theresa Feltwell[43], Anita Kenyon[44], Sahar Eldirdiri[44], Thomas Davis[44], Joshua F. Taylor[45], Ngee Keong Tan[45], Alex E. Zarebski[46], Bernardo Gutierrez[46], Jayna Raghwani[46], Louis du Plessis[46], Moritz U. G. Kraemer[46], Oliver G. Pybus[46], Sarah Francois[46], Stephen W. Attwood[46], Tetyana I. Vasylyeva[46], Aminu S. Jahun[47], Ian G. Goodfellow[47], Iliana Georgana[47], Malte L. Pinckert[47], Myra Hosmillo[47], Rhys Izuagbe[47], Yasmin Chaudhry[47], Felicity Ryan[48], Hannah Lowe[48], Samuel Moses[48], Luke Bedford[49], James S. Cargill[50], Warwick Hughes[50], Jonathan Moore[51],**

Susanne Stonehouse[51], Divya Shah[52], Jack C. D. Lee[52], Julianne R. Brown[52], Kathryn A. Harris[52], Laura Atkinson[52], Nathaniel Storey[52], Moira J. Spyer[53], Flavia Flaviani[54], Adela Alcolea-Medina[55], Jasveen Sehmi[55], John Ramble[55], Natasha Ohemeng-Kumi[55], Perminder Smith[55], Beatrice Bertolusso[56], Claire Thomas[56], Gabrielle Vernet[56], Jessica Lynch[56], Nathan Moore[56], Nicholas Cortes[56], Rebecca Williams[56], Stephen P. Kidd[56], Lisa J. Levett[57], Monika Pusok[57], Paul R. Grant[57], Stuart Kirk[57], Wendy Chatterton[57], Li Xu-McCrae[58], Darren L. Smith[59], Gregory R. Young[59], Matthew Bashton[59], Katie Kitchman[60], Kavitha Gajee[60], Kirstine Eastick[60], Patrick J. Lillie[60], Phillipa J. Burns[60], William Everson[60], Alison Cox[61], Alison H. Holmes[61], Frances Bolt[61], James R. Price[61], Marcus Pond[61], Paul A. Randell[61], Pinglawathee Madona[61], Siddharth Mookerjee[61], Erik M. Volz[62], Lily Geidelberg[62], Manon Ragonnet-Cronin[62], Olivia Boyd[62], Rob Johnson[62], Cassie F. Pope[63], Adam A. Witney[64], Irene M. Monahan[64], Kenneth G. Laing[64], Katherine L. Smollett[65], Alan McNally[66], Claire McMurray[66], Joanne Stockton[66], Joshua Quick[66], Nicholas J. Loman[66], Radoslaw Poplawski[66], Sam Nicholls[66], Will Rowe[66], Anibolina Castigador[67], Emily Macnaughton[67], Kate El Bouzidi[68], Malur Sudhanva[68], Temi Lampejo[68], Rocio T. Martinez Nunez[69], Cassie Breen[70], Graciela Sluga[71], Karen T. Withell[71], Nicholas W. Machin[72], Ryan P. George[72], Shazaad S. Y. Ahmad[72], David T. Pritchard[73], Debbie Binns[73], Nick Wong[73], Victoria James[74], Cheryl Williams[75], Chris J. Illingworth[76], Chris Jackson[76], Daniela de Angelis[76], David Pascall[76], Afrida Mukaddas[77], Alice Broos[77], Ana da Silva Filipe[77], Daniel Mair[77], David L. Robertson[77], Derek W. Wright[77], Emma C. Thomson[77], Igor Starinskij[77], Ioulia Tsatsani[77], James G. Shepherd[77], Jenna Nichols[77], Joseph Hughes[77], Kyriaki Nomikou[77], Lily Tong[77], Richard J. Orton[77], Sreenu Vattipally[77], William T. Harvey[77], Roy Sanderson[78], Sarah O'Brien[78], Steven Rushton[78], Jon Perkins[79], Rachel Blacow[79], Rory N. Gunson[79], Abbie Gallagher[80], Elizabeth Wastnedge[80], Kate E. Templeton[80], Martin P. McHugh[80], Rebecca Dewar[80], Seb Cotton[80], Lindsay Coupland[81], Rachael Stanley[81], Samir Dervisevic[81], Lewis G. Spurgin[82], Louise Smith[82], Clive Graham[83], Debra Padgett[83], Edward Barton[83], Garren Scott[83], Aidan Cross[84], Mariyam Mirfenderesky[84], Emma Swindells[85], Jane Greenaway[85], Rebecca Denton-Smith[85], Robyn Turnbull[85], Giles Idle[86], Kevin Cole[86], Amy Hollis[87], Andrew Nelson[87], Clare M. McCann[87], John H. Henderson[87], Matthew R. Crown[87], Wen C. Yew[87], William Stanley[87], Nichola Duckworth[88], Phillip Clarke[88], Sarah Walsh[88], Tim J. Sloan[88], Kelly Bicknell[89], Robert Impey[89], Sarah Wyllie[89], Scott Elliott[89], Sharon Glaysher[89], Declan T. Bradley[90], Nicholas F. Killough[90], Tim Wyatt[90], Andrew Bosworth[91], Barry B. Vipond[91], Clare Pearson[91], Elias Allara[91], Esther Robinson[91], Hannah M. Pymont[91], Husam Osman[91], Peter Muir[91], Richard Hopes[91], Stephanie Hutchings[91], Martin D. Curran[92], Surendra Parmar[92], Alicia Thornton[93], Angie Lackenby[93], Chloe Bishop[93], David Bibby[93], David Lee[93], Eileen Gallagher[93], Gavin Dabrera[93], Ian Harrison[93], Jonathan Hubb[93], Katherine A. Twohig[93], Meera Chand[93], Nicholas Ellaby[93], Nikos Manesis[93], Richard Myers[93], Steven Platt[93], Tamyo Mbisa[93], Vicki Chalker[93], Gonzalo Yebra[94], Matthew T. G. Holden[94], Sharif Shaaban[94], Stefan Rooke[94], Alec Birchley[95], Alexander Adams[95], Alisha Davies[95], Amy Gaskin[95], Bree Gatica-Wilcox[95], Caoimhe McKerr[95], Catherine Moore[95], Catryn Williams[95], Chris Williams[95], David Heyburn[95], Elen De Lacy[95], Ember Hilvers[95], Fatima Downing[95], Georgia Pugh[95], Hannah Jones[95], Hibo Asad[95], Jason Coombes[95], Jessica Hey[95], Jessica Powell[95], Joanne Watkins[95], Johnathan M. Evans[95], Laia Fina[95], Laura Gifford[95], Lauren Gilbert[95], Lee Graham[95], Malorie Perry[95], Mari Morgan[95], Matthew Bull[95], Nicole Pacchiarini[95], Noel Craine[95], Sally Corden[95], Sara Kumziene-Summerhayes[95], Sara Rey[95], Sarah Taylor[95], Simon Cottrell[95], Sophie Jones[95],

Sue Edwards[95], Tara Annett[95], Alexander J. Trotter[96], Alison E. Mather[96], Alp Aydin[96], Andrew J. Page[96], David J. Baker[96], Ebenezer Foster-Nyarko[96], Gemma L. Kay[96], Justin O'Grady[96], Leonardo de Oliveira Martins[96], Lizzie Meadows[96], Nabil-Fareed Alikhan[96], Sophie J. Prosolek[96], Steven Rudder[96], Thanh Le-Viet[96], Anna Casey[97], Liz Ratcliffe[97], Aditi Singh[98], Arun Mariappan[98], Chris Baxter[98], Clara Radulescu[98], David A. Simpson[98], Deborah Lavin[98], Fiona Rogan[98], Julia Miskelly[98], Marc Fuchs[98], Miao Tang[98], Sílvia F. Carvalho[98], Stephen Bridgett[98], Timofey Skvortsov[98], Zoltan Molnar[98], Newara A. Ramadan[99], Bridget A. Knight[100], Christopher R. Jones[100], Cressida Auckland[100], Helen Morcrette[100], Jennifer Poyner[100], Dianne Irish-Tavares[101], Eric Witele[101], Jennifer Hart[101], Tabitha W. Mahungu[101], Tanzina Haque[101], Yann Bourgeois[102], Christopher Fearn[103], Kate F. Cook[103], Katie F. Loveson[103], Salman Goudarzi[103], Cariad Evans[104], David G. Partridge[104], Kate Johnson[104], Mehmet Yavus[104], Mohammad Raza[104], Craig Mower[105], Paul Baker[105], Sarah Essex[105], Stephen Bonner[105], Leanne J. Murray[105], Louisa K. Watson[105], Steven Liggett[105], Andrew I. Lawton[106], Ronan A. Lyons[107], Brendan A. I. Payne[108], Gary Eltringham[108], Jennifer Collins[108], Sheila Waugh[108], Shirelle Burton-Fanning[108], Yusri Taha[108], Christopher Jeanes[109], Andrea N. Gomes[110], Darren R. Murray[110], Maimuna Kimuli[110], Donald Dobie[111], Paula Ashfield[111], Angus Best[112], Benita Percival[112], Emma Moles-Garcia[112], Fiona Ashford[112], Jeremy Mirza[112], Liam Crawford[112], Megan Mayhew[112], Nicola Cumley[112], Oliver Megram[112], Dan Frampton[113], Judith Heaney[114], Matthew Byott[114], Catherine Houlihan[115], Charlotte A. Williams[115], Eleni Nastouli[115], Helen L. Lowe[115], John A. Hartley[115], Judith Breuer[115], Laurentiu Maftei[115], Leah Ensell[115], Marius Cotic[115], Matteo Mondani[115], Megan Driscoll[115], Nadua Bayzid[115], Rachel J. Williams[115], Sunando Roy[115], Adhyana I. K. Mahanama[116], Buddhini Samaraweera[116], Eleri Wilson-Davies[116], Emanuela Pelosi[116], Helen Umpleby[116], Helen Wheeler[116], Jacqui A. Prieto[116], Kordo Saeed[116], Matthew Harvey[116], Sarah Jeremiah[116], Siona Silviera[116], Stephen Aplin[116], Thea Sass[116], Ben Macklin[117], Dorian Crudgington[117], Liz A. Sheridan[117], Benjamin J. Cogger[118], Cassandra S. Malone[118], Florence Munemo[118], Hannah Huckson[118], Jonathan Lewis[118], Lisa J. Easton[118], Manasa Mutingwende[118], Michelle J. Erkiert[118], Mohammed O. Hassan-Ibrahim[118], Nicola J. Chaloner[118], Olga Podplomyk[118], Paul Randell[118], Roberto Nicodemi[118], Sarah Lowdon[118], Thomas Somassa[118], Alex Richter[119], Andrew Beggs[119], Andrew R. Hesketh[120], Colin P. Smith[120], Giselda Bucca[120], Chris Ruis[121], Claire Cormie[121], Ellen E. Higginson[121], Jamie Young[121], Joana Dias[121], Leanne M. Kermack[121], Mailis Maes[121], Ravi K. Gupta[121], Sally Forrest[121], Sophia T. Girgis[121], Rose K. Davidson[122], Áine O'Toole[123], Andrew Rambaut[123], Ben Jackson[123], Carlos E. Balcazar[123], Daniel Maloney[123], Emily Scher[123], J. T. McCrone[123], Kathleen A. Williamson[123], Michael D. Gallagher[123], Nathan Medd[123], Rachel Colquhoun[123], Thomas D. Stanton[123], Thomas Williams[123], Verity Hill[123], Aaron R. Jeffries[124], Ben Temperton[124], Christine M. Sambles[124], David J. Studholme[124], Joanna Warwick-Dugdale[124], Leigh M. Jackson[124], Michelle L. Michelsen[124], Robin Manley[124], Stephen L. Michell[124], Alistair C. Darby[125], Anita O. Lucaci[125], Charlotte Nelson[125], Claudia Wierzbicki[125], Edith E. Vamos[125], Hermione J. Webster[125], Kathryn A. Jackson[125], Lucille Rainbow[125], Margaret Hughes[125], Mark Whitehead[125], Matthew Gemmell[125], Miren Iturriza-Gomara[125], Richard Eccles[125], Richard Gregory[125], Sam T. Haldenby[125], Steve Paterson[125], Adrienn Angyal[126], Alexander J. Keeley[126], Benjamin H. Foulkes[126], Benjamin B. Lindsey[126], Dennis Wang[126], Hailey R. Hornsby[126], Luke R. Green[126], Manoj Pohare[126], Marta Gallis[126], Matthew D. Parker[126], Max Whiteley[126], Nikki Smith[126], Paige Wolverson[126], Peijun Zhang[126], Samantha E. Hansford[126], Sharon N. Hsu[126], Stavroula F. Louka[126], Thushan I. de Silva[126], Timothy M. Freeman[126], Matilde Mori[127], Emily J. Park[128], Jack D. Hill[128],

Jayasree Dey[128], Jonathan Ball[128], Joseph G. Chappell[128], Patrick C. McClure[128], Timothy Byaruhanga[128], Arezou Fanaie[129], Geraldine Yaze[129], Rachel A. Hilson[129], Amy Trebes[130], Angie Green[130], David Buck[130], George MacIntyre-Cockett[130], John A. Todd[130], Andrew R. Bassett[131], Andrew Whitwham[131], Cordelia F. Langford[131], Diana Rajan[131], Dominic Kwiatkowski[131], Ewan M. Harrison[131], Iraad F. Bronner[131], Jaime M. Tovar-Corona[131], Jennifier Liddle[131], Jillian Durham[131], Katherine L. Bellis[131], Kevin Lewis[131], Louise Aigrain[131], Nicholas M. Redshaw[131], Robert M. Davies[131], Robin J. Moll[131], Shane A. McCarthy[131], Stefanie V. Lensing[131], Steven Leonard[131], Ben W. Farr[131], Carol Scott[131], Charlotte Beaver[131], Cristina V. Ariani[131], Danni Weldon[131], David K. Jackson[131], Emma Betteridge[131], Gerry Tonkin-Hill[131], Ian Johnston[131], Inigo Martincorena[131], James Bonfield[131], Jeffrey C. Barrett[131], John Sillitoe[131], Jon-Paul Keatley[131], Karen Oliver[131], Keith James[131], Lesley Shirley[131], Liam Prestwood[131], Luke Foulser[131], Marina Gourtovaia[131], Matthew J. Dorman[131], Michael A. Quail[131], Michael H. Spencer Chapman[131], Naomi R. Park[131], Rich Livett[131], Roberto Amato[131], Sally Kay[131], Scott Goodwin[131], Scott A. J. Thurston[131], Shavanthi Rajatileka[131], Sónia Gonçalves[131], Stephanie Lo[131], Theo Sanderson[131], Alasdair Maclean[132], Emily J. Goldstein[132], Lynne Ferguson[132], Rachael Tomb[132], Jana Catalan[133] and Neil Jones[133]

[24]Barking, Havering and Redbridge University Hospitals NHS Trust, Romford, UK. [25]Barts Health NHS Trust, London, UK. [26]Belfast Health and Social Care Trust, Belfast, UK. [27]Betsi Cadwaladr University Health Board, Bangor, UK. [28]Big Data Institute, Nuffield Department of Medicine, University of Oxford, Oxford, UK. [29]Blackpool Teaching Hospitals NHS Foundation Trust, Blackpool, UK. [30]Bournemouth University, Bournemouth, UK. [31]Cambridge Stem Cell Institute, University of Cambridge, Cambridge, UK. [32]Cambridge University Hospitals NHS Foundation Trust, Cambridge, UK. [33]Cardiff and Vale University Health Board, Cardiff, UK. [34]Cardiff University, Cardiff, UK. [35]Centre for Clinical Infection and Diagnostics Research, Department of Infectious Diseases, Guy's and St Thomas' NHS Foundation Trust, London, UK. [36]Centre for Enzyme Innovation, University of Portsmouth, Portsmouth, UK. [37]Centre for Genomic Pathogen Surveillance, University of Oxford, Oxford, UK. [38]Clinical Microbiology Department, Queens Medical Centre, Nottingham University Hospitals NHS Trust, Nottingham, UK. [39]Clinical Microbiology, University Hospitals of Leicester NHS Trust, Leicester, UK. [40]County Durham and Darlington NHS Foundation Trust, Darlington, UK. [41]Deep Seq, School of Life Sciences, Queens Medical Centre, University of Nottingham, Nottingham, UK. [42]Department of Infectious Diseases and Microbiology, Cambridge University Hospitals NHS Foundation Trust, Cambridge, UK. [43]Department of Medicine, University of Cambridge, Cambridge, UK. [44]Department of Microbiology, Kettering General Hospital, Kettering, UK. [45]Department of Microbiology, South West London Pathology, London, UK. [46]Department of Zoology, University of Oxford, Oxford, UK. [47]Division of Virology, Department of Pathology, University of Cambridge, Cambridge, UK. [48]East Kent Hospitals University NHS Foundation Trust, Canterbury, UK. [49]East Suffolk and North Essex NHS Foundation Trust, Colchester, UK. [50]East Sussex Healthcare NHS Trust, St. Leonards-on-Sea, UK. [51]Gateshead Health NHS Foundation Trust, Gateshead, UK. [52]Great Ormond Street Hospital for Children NHS Foundation Trust, London, UK. [53]Great Ormond Street Institute of Child Health (GOS ICH), University College London (UCL), London, UK. [54]Guy's and St. Thomas' Biomedical Research Centre, London, UK. [55]Guy's and St. Thomas' NHS Foundation Trust, London, UK. [56]Hampshire Hospitals NHS Foundation Trust, Basingstoke, UK. [57]Health Services Laboratories, London, UK. [58]Heartlands Hospital, Birmingham, Birmingham, UK. [59]Hub for Biotechnology in the Built Environment, Northumbria University, Newcastle upon Tyne, UK. [60]Hull University Teaching Hospitals NHS Trust, Hull, UK. [61]Imperial College Healthcare NHS Trust, London, UK. [62]Imperial College London, London, UK. [63]Infection Care Group, St George's University Hospitals NHS Foundation Trust, London, UK. [64]Institute for Infection and Immunity, St George's University of London, London, UK. [65]Institute of Biodiversity, Animal Health & Comparative Medicine, University of Glasgow, Glasgow, UK. [66]Institute of Microbiology and Infection, University of Birmingham, Birmingham, UK. [67]Isle of Wight NHS Trust, Newport, UK. [68]King's College Hospital NHS Foundation Trust, London, UK. [69]King's College London, London, UK. [70]Liverpool Clinical Laboratories, Liverpool, UK. [71]Maidstone and Tunbridge Wells NHS Trust, Maidstone, UK. [72]Manchester University NHS Foundation Trust, Manchester, UK. [73]Microbiology Department, Buckinghamshire Healthcare NHS Trust, Aylesbury, UK. [74]Microbiology Department, Buckinghamshire Healthcare NHS Trust, Aylesbury, UK. [75]Microbiology, Royal Oldham Hospital, Oldham, UK. [76]MRC Biostatistics Unit, University of Cambridge, Cambridge, UK. [77]MRC-University of Glasgow Centre for Virus Research, Glasgow, UK. [78]Newcastle University, Newcastle upon Tyne, UK. [79]NHS Greater Glasgow and Clyde, Glasgow, UK. [80]NHS Lothian, Edinburgh, UK. [81]Norfolk and Norwich University Hospitals NHS Foundation Trust, Norwich, UK. [82]Norfolk County Council, Norwich, UK. [83]North Cumbria Integrated Care NHS Foundation Trust, Carlisle, UK. [84]North Middlesex University Hospital NHS Trust, London, UK. [85]North Tees and Hartlepool NHS Foundation Trust, Stockton on Tees, UK. [86]Northumbria Healthcare NHS Foundation Trust, Newcastle upon Tyne, UK. [87]Northumbria University, Newcastle upon Tyne, UK. [88]Path Links, Northern Lincolnshire and Goole NHS Foundation Trust, Grimsby, UK. [89]Portsmouth Hospitals University NHS Trust, Portsmouth, UK. [90]Public Health Agency, Northern Ireland, Belfast, UK. [91]Public Health England, London, UK. [92]Public Health England, Cambridge, Cambridge, UK. [93]Public Health England, Colindale, London, UK. [94]Public Health Scotland, Edinburgh, UK. [95]Public Health Wales, Cardiff, UK. [96]Quadram Institute Bioscience, Norwich, UK. [97]Queen Elizabeth Hospital, Birmingham, Birmingham, UK. [98]Queen's University Belfast, Belfast, UK. [99]Royal Brompton and Harefield Hospitals, London, UK. [100]Royal Devon and Exeter NHS Foundation Trust, Exeter, UK. [101]Royal Free London NHS Foundation Trust, London, UK. [102]School of Biological Sciences, University of Portsmouth, Portsmouth, UK. [103]School of Pharmacy & Biomedical Sciences, University of Portsmouth, Portsmouth, UK. [104]Sheffield Teaching Hospitals NHS Foundation Trust, Sheffield, UK. [105]South Tees Hospitals NHS Foundation Trust, Middlesbrough, UK. [106]Southwest Pathology Services, Taunton, UK. [107]Swansea University, Swansea, UK. [108]The Newcastle upon Tyne Hospitals NHS Foundation Trust, Newcastle upon Tyne, UK. [109]The Queen Elizabeth Hospital King's Lynn NHS Foundation Trust, King's Lynn, UK. [110]The Royal Marsden NHS Foundation Trust, London, UK. [111]The Royal Wolverhampton NHS Trust, Wolverhampton, UK. [112]Turnkey Laboratory, University of Birmingham, Birmingham, UK. [113]University College London Division of Infection and Immunity, London, UK. [114]University College London Hospital Advanced Pathogen Diagnostics Unit, London, UK. [115]University College London Hospitals NHS Foundation Trust, London, UK. [116]University Hospital Southampton NHS Foundation Trust, Southampton, UK. [117]University Hospitals Dorset NHS Foundation Trust, Poole, UK. [118]University Hospitals Sussex NHS Foundation Trust, Worthing, UK. [119]University of Birmingham, Birmingham, UK. [120]University of Brighton, Brighton, UK. [121]University of Cambridge, Cambridge, UK. [122]University of East Anglia, Norwich, UK. [123]University of Edinburgh, Edinburgh, UK. [124]University of Exeter, Exeter, UK. [125]University of Liverpool, Liverpool, UK. [126]University of Sheffield, Sheffield, UK. [127]University of

Southampton, Nottingham, UK. [128]Virology, School of Life Sciences, Queens Medical Centre, University of Nottingham, Nottingham, UK. [129]Watford General Hospital, Watford, UK. [130]Wellcome Centre for Human Genetics, Nuffield Department of Medicine, University of Oxford, Oxford, UK. [131]Wellcome Sanger Institute, Hinxton, UK. [132]West of Scotland Specialist Virology Centre, NHS Greater Glasgow and Clyde, Glasgow, UK. [133]Whittington Health NHS Trust, London, UK.

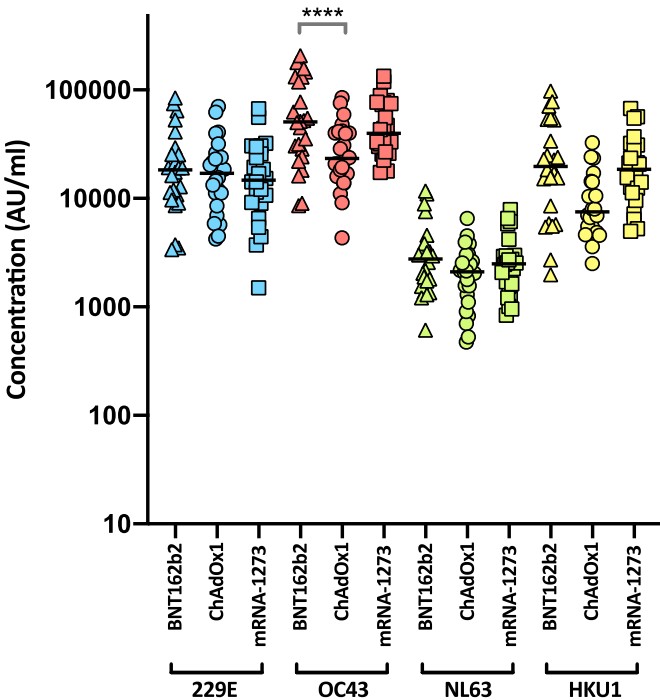

**Extended Data Fig. 1 | HCoV reactivity following two doses of SARS-CoV-2 vaccine.** Antibody responses were studied in three groups of individuals (n = 24 per group) vaccinated with either BNT162b2, ChAdOx1 or mRNA-1273 by MSD-ECL assay. Responses were measured against full-length spike glycoprotein (Spike) from HCoVs 229E, OC43, NL63 and HKU1 and are expressed as MSD arbitrary units (AU/ml). The response to OC43 was significantly higher in BNT162b2 vaccinates than in ChAdOx1 vaccinates. Bar represents group mean. Group means compared by one-way ANOVA, Tukey's multiple comparisons test, **** significantly different $p < 0.0001$.

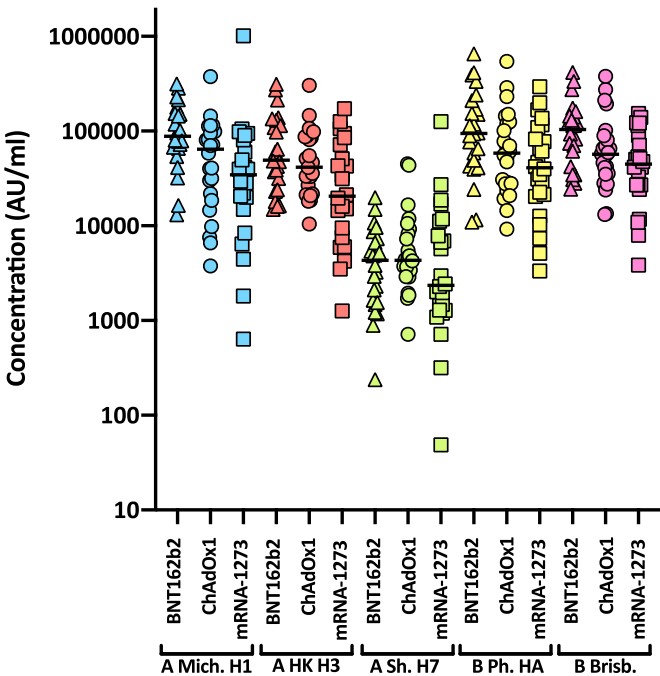

**Extended Data Fig. 2 | Influenza reactivity following two doses of SARS-CoV-2 vaccine.** Antibody responses were studied in three groups of individuals (n = 24 per group) vaccinated with either BNT162b2, ChAdOx1 or mRNA-1273 by MSD-ECL assay. Responses were measured against haemagglutinins from influenza viruses; influenza A Michigan H1, Hong Kong H3 and Shanghai H7, and influenza B Phuket HA and Brisbane and are expressed as MSD arbitrary units (AU/ml). No significant differences were detected between the vaccine groups for each of the antigens. Bar represents group mean.

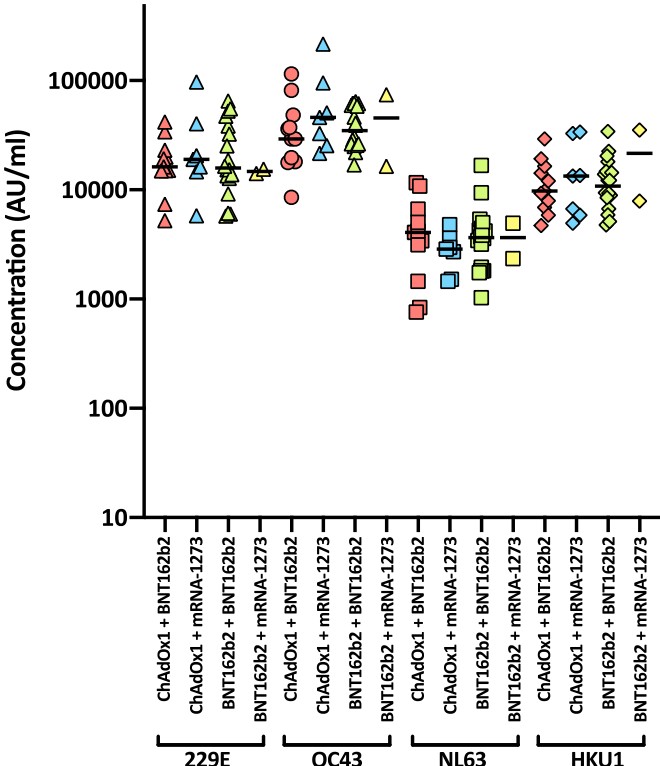

**Extended Data Fig. 3 | HCoV reactivity following third dose of SARS-CoV-2 vaccine.** Antibody responses were studied in four groups of individuals primed with two doses of either ChAdOx1 or BNT162b2, followed by a booster of BNT162b2 or mRNA-1273. Responses were measured by MSD-ECL assay against full-length spike glycoprotein (Spike) from HCoVs 229E, OC43, NL63 and HKU1 and are expressed as MSD arbitrary units (AU/ml). Bar represents group mean.

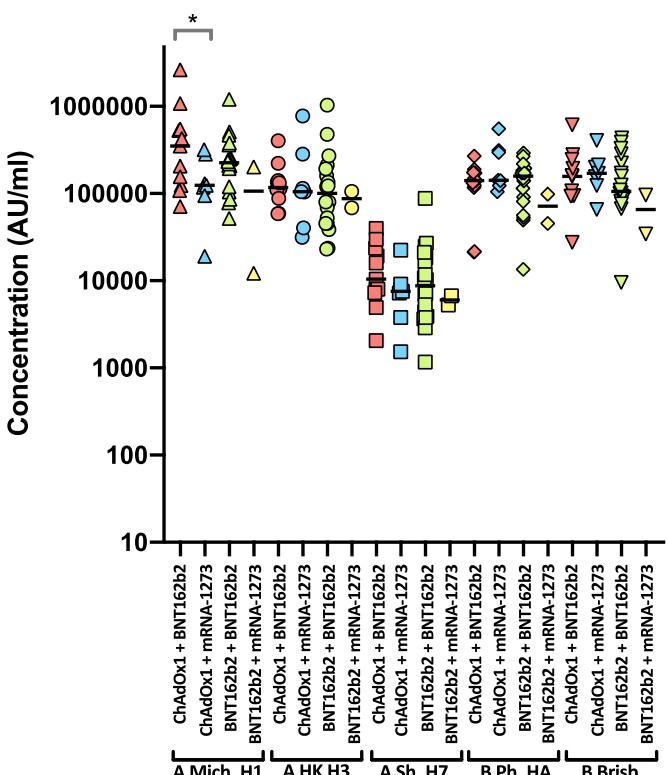

**Extended Data Fig. 4 | Influenza reactivity following third dose of SARS-CoV-2 vaccine.** Antibody responses were studied in four groups of individuals primed with two doses of either ChAdOx1 or BNT162b2, followed by a booster of BNT162b2 or mRNA-1273. Responses were measured by MSD-ECL against haemagglutinins from influenza viruses; influenza A Michigan H1, Hong Kong H3 and Shanghai H7, and influenza B Phuket HA and Brisbane and are expressed as MSD arbitrary units (AU/ml). Bar represents group mean. Group means compared by one-way ANOVA, Tukey's multiple comparisons test, * significantly different p = 0.0413.

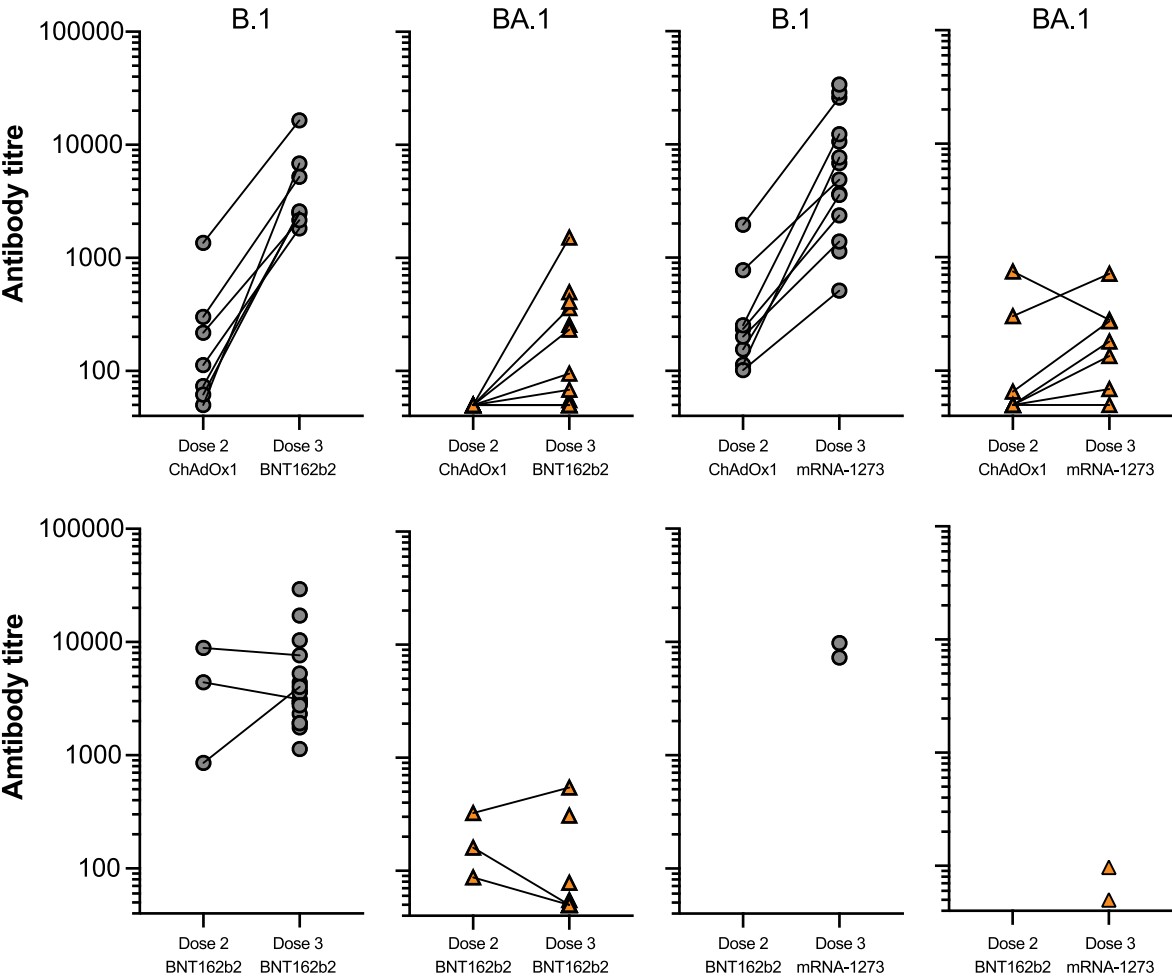

**Extended Data Fig. 5 | Effect of third dose of SARS-CoV-2 vaccine on neutralising antibody titres.** Two groups of healthy volunteers vaccinated with two doses of either ChAdOx1 or BNT162b2, were sampled two weeks following a third dose of either BNT162b2 or mRNA-1273. Each point represents the mean of three replicates. Where dose 2 and dose 3 samples were available from the same individual, points are joined by a solid line.

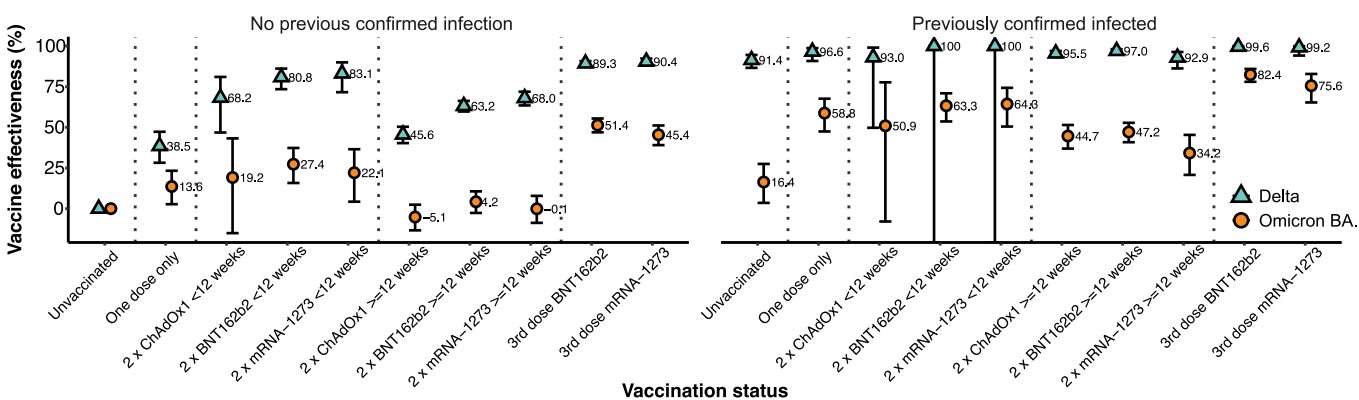

**Extended Data Fig. 6 | Vaccine effectiveness estimates considering time since second dose.** Error bar plot of estimated vaccine effectiveness against testing positive for Delta and Omicron SARS-CoV-2 infection in the population of over 18 s in NHS GG&C who were tested between 6th and 26th December 2021. The points and corresponding text represent the estimated vaccine effectiveness (%) for each group, for each variant, with the error bar endpoints representing the endpoints of the corresponding 95% CIs. The modelling process was identical to the main vaccine effectiveness estimation reported in the main document, but the vaccine status variable had additional levels for 2nd dose within 12 weeks of start of study period or before 12 weeks of start of study period. Note that the estimates for infection with Delta after previous confirmed SARS-CoV-2 infection and 2 x BNT162b2 < 12 weeks and 2 x mRNA-1272 < 12 weeks are calculated for a group with no infections and are therefore unreliable.

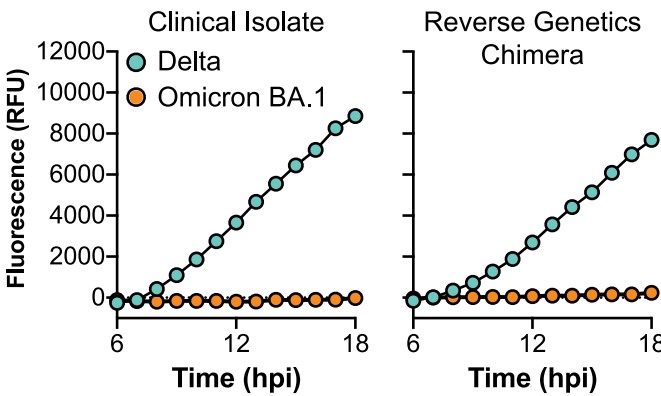

**Extended Data Fig. 7 | Comparison of syncytia formation by clinical isolates and reverse-genetics live virus.** GFP-10 and GFP-11 A549 ACE2 TMPRSS2 cells were co-cultured and infected with Delta and Omicron BA.1 clinical isolates or with live reverse-genetics virus in which the Delta or Omicron BA.1 spike is presented in the context of the ancestral wildtype B lineage genome. Fusion was quantified by measuring GFP signal, as in Fig. 5.

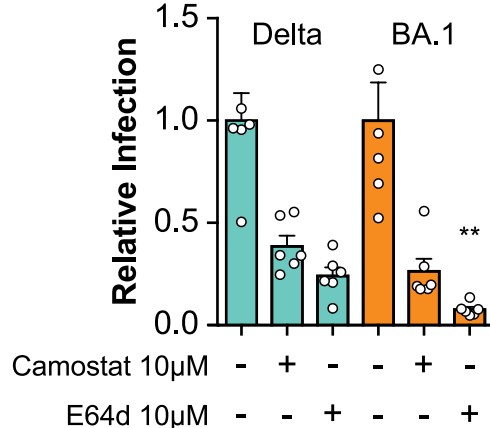

**Extended Data Fig. 8 | Sensitivity of live SARS-CoV-2 to protease inhibitors.** SARS-CoV-2 infection of A549 ACE2 TMPRSS2 cells in the presence of 10 μM Camostat or E64d, data is expressed relative to untreated control, values represent mean across two independent experiments, asterisks indicate statistical significance (Two tailed T-test) between E64d treated Delta and Omicron infections. Error bars indicate standard error of the mean.

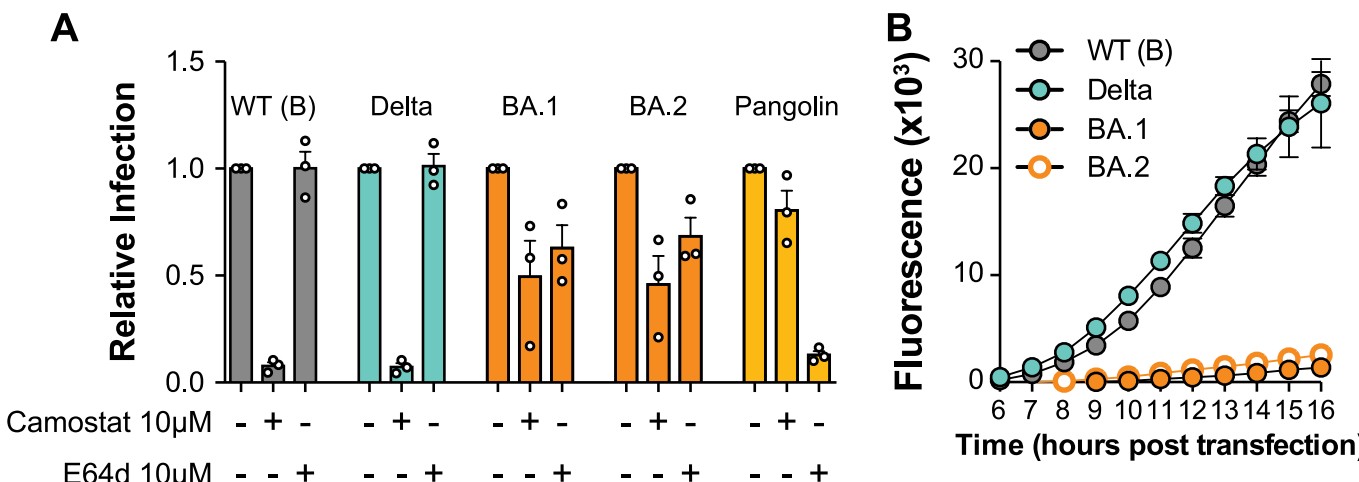

**Extended Data Fig. 9 | Characterisation of Omicron BA.2 spike. a**, Relative SARS-CoV-2 pseudotype infection (compared to respective untreated controls) treated with 10 µM protease inhibitors. Data represent mean of three biological repeats, error bars indicate standard error of the mean. **b**, cell fusion assay using cells transfected to express WT (B lineage), Delta, Omicron BA.1 and BA.2 spike, values represent mean GFP fluorescence signal from one representative experiment error bars indicate standard error of the mean (3 technical repeats, representative of 4 biological repeats).

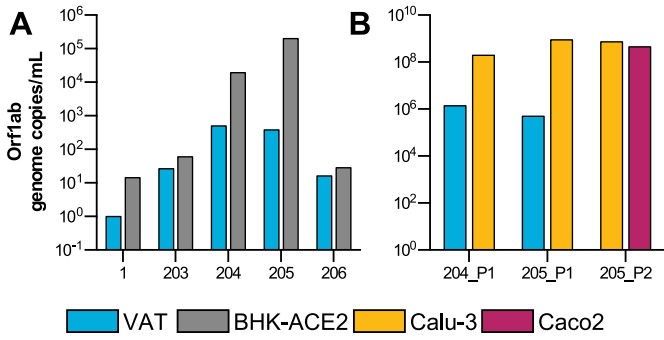

**Extended Data Fig. 10 | Isolation of Omicron in cell culture. a**, Vero ACE2 TMPRSS2 (VAT) and BHK-hACE2 cells were inoculated with diluted clinical samples. Viral progeny was quantified in the medium 5 dpi by RT-qPCR. **b**, Aliquots of the medium from samples named 204 and 205 were used to generate a P1 in BHK-hACE2 and Calu-3 cells and, limited to sample 205, a P2 in Calu-3 and Caco2 cells. Viral stocks were quantified by RT-qPCR.

| | |
|---|---|

# Reporting Summary

## Statistics

For all statistical analyses, confirm that the following items are present in the figure legend, table legend, main text, or Methods section.

| n/a | Confirmed | |
|---|---|---|
| ☐ | ☒ | The exact sample size ($n$) for each experimental group/condition, given as a discrete number and unit of measurement |
| ☐ | ☒ | A statement on whether measurements were taken from distinct samples or whether the same sample was measured repeatedly |
| ☐ | ☒ | The statistical test(s) used AND whether they are one- or two-sided<br>*Only common tests should be described solely by name; describe more complex techniques in the Methods section.* |
| ☐ | ☒ | A description of all covariates tested |
| ☐ | ☒ | A description of any assumptions or corrections, such as tests of normality and adjustment for multiple comparisons |
| ☐ | ☒ | A full description of the statistical parameters including central tendency (e.g. means) or other basic estimates (e.g. regression coefficient) AND variation (e.g. standard deviation) or associated estimates of uncertainty (e.g. confidence intervals) |
| ☐ | ☒ | For null hypothesis testing, the test statistic (e.g. $F$, $t$, $r$) with confidence intervals, effect sizes, degrees of freedom and $P$ value noted<br>*Give P values as exact values whenever suitable.* |
| ☒ | ☐ | For Bayesian analysis, information on the choice of priors and Markov chain Monte Carlo settings |
| ☒ | ☐ | For hierarchical and complex designs, identification of the appropriate level for tests and full reporting of outcomes |
| ☒ | ☐ | Estimates of effect sizes (e.g. Cohen's $d$, Pearson's $r$), indicating how they were calculated |

*Our web collection on statistics for biologists contains articles on many of the points above.*

## Software and code

Policy information about availability of computer code

| Data collection | Anonymised data were made available for analysis on the NHS GGC SafeHaven platform (EVADE) |
|---|---|
| Data analysis | Statistical analysis for the vaccine effectiveness calculations was carried out in R version 4.0.5 on the NHS GGC SafeHaven platform. The R scripts are available in the GitHub repository (https://github.com/centre-for-virus-research/Omicron). |

For manuscripts utilizing custom algorithms or software that are central to the research but not yet described in published literature, software must be made available to editors and reviewers. We strongly encourage code deposition in a community repository (e.g. GitHub). See the Nature Portfolio guidelines for submitting code & software for further information.

## Data

Policy information about availability of data

All manuscripts must include a data availability statement. This statement should provide the following information, where applicable:
- Accession codes, unique identifiers, or web links for publicly available datasets
- A description of any restrictions on data availability
- For clinical datasets or third party data, please ensure that the statement adheres to our policy

The experimental data that support the findings of this study are included with the submission (neutralisation, ELISpot, entry data) but restrictions apply to the availability of clinical data, which were used under ethical approvals for the current study behind an NHS firewall, and so are not publicly available. Biological materials including cell lines are available on reasonable request from the authors. Clinical samples are restricted for use under the ethical approvals obtained for their use.

# Field-specific reporting

Please select the one below that is the best fit for your research. If you are not sure, read the appropriate sections before making your selection.

☒ Life sciences        ☐ Behavioural & social sciences        ☐ Ecological, evolutionary & environmental sciences

For a reference copy of the document with all sections, see nature.com/documents/nr-reporting-summary-flat.pdf

# Life sciences study design

All studies must disclose on these points even when the disclosure is negative.

| | |
|---|---|
| Sample size | For evaluation of vaccine effectiveness, all data points for people over 18 years old and living in the NHS GGC area with a PCR test for COVID-19 carried out between 2021-12-06 and 2021-12-26 were included in the analysis (and with either sequencing information, ASP status or SGTF status) recorded, for positives). For neutralisation (DOVE), age-matched participants (24/group) were selected. |
| Data exclusions | Those with a vaccine listed other than ChAdOxl, BNT162b2 or mRNA-1273, and those with multiple vaccinations listed on the same day with different products, were removed from the study. Only those with a PCR test for COVID-19 carried out between 2021-12-06 and 2021-12-26 (and with either sequencing information, ASP status or SGTF status recorded, for positives) were included in the study, to avoid biases due to incorrect population baseline estimates. Those who were due to receive a new vaccine dose but did not were excluded, to avoid bias due to infection delaying vaccination. Those who received a new vaccine dose during the study period were also excluded. |
| Replication | In vitro studies were performed using multiple replicates, the number of which is specified in each independent figure, source data are provided separately. |
| Randomization | Allocation to case or control group was defined by COVID-19 PCR status. This non-random allocation was controlled for using data on age, sex and deprivation index (SIMD) quartile for each participant (for calculation of vaccine effectiveness.) |
| Blinding | Blinding was not carried out, due to the observational nature of the study, with no randomization required or possible. |

# Reporting for specific materials, systems and methods

We require information from authors about some types of materials, experimental systems and methods used in many studies. Here, indicate whether each material, system or method listed is relevant to your study. If you are not sure if a list item applies to your research, read the appropriate section before selecting a response.

## Materials & experimental systems

| n/a | Involved in the study |
|---|---|
| ☐ | ☒ Antibodies |
| ☐ | ☒ Eukaryotic cell lines |
| ☒ | ☐ Palaeontology and archaeology |
| ☒ | ☐ Animals and other organisms |
| ☐ | ☒ Human research participants |
| ☒ | ☐ Clinical data |
| ☒ | ☐ Dual use research of concern |

## Methods

| n/a | Involved in the study |
|---|---|
| ☒ | ☐ ChIP-seq |
| ☒ | ☐ Flow cytometry |
| ☒ | ☐ MRI-based neuroimaging |

## Antibodies

| | |
|---|---|
| Antibodies used | The plates were incubated for 18 hours at 37oC, 5% CO2. The plates were then washed with PBS and incubated with 1µg/ml biotin-labelled detection antibody (7-B6-1, Mabtech) |
| Validation | Documentation and validation of this commercial antibody can be found at: https://www.mabtech.com/products/anti-human-ifn-gamma-antibody-7-b6-1-biotinylated-3420-6 |

## Eukaryotic cell lines

Policy information about cell lines

| | |
|---|---|
| Cell line source(s) | Cells. Calu-3 cells ATCC #HTB-55 are human lung adenocarcinoma epithelial cells. Caco-2 (CVR cytology cell bank) are an immortalized cell line derived from human colorectal adenocarcinoma, primarily used as a model of the intestinal epithelial barrier. A549 cells (ATCC #CCL-185) a human alveolar adenocarcinoma line, were a generous gift from Prof. Ben Hale, validated by STR analysis (Eurofins)). A549 were modified to stably express human ACE-2 and TMPRSS2. Human embryonic kidney (HEK293T) cells were used in pseudotype production. African green monkey kidney cells (Vero) were used to |

propagate the reverse genetics-derived viruses. Baby Hamster Kidney clone 21 cells (BHK-21 ATCC #CCL-10, purchased from ATCC, Bethesda, MD) and Vero ACE-2 TMPRSS272 cells were used in the isolation of live Omicron SARS-CoV-2. All cell lines were maintained at 37°C and 5% $CO_2$ in DMEM supplemented with 10% foetal bovine serum (FBS), except for Calu-3 cells which were supplemented with 20% FBS. Human reconstituted upper airway epithelium (Mucilair™, abbreviated hNECs in this manuscript) were purchased from Epithelix and maintained in Mucilair complete culture basal medium (Epithelix) at an air-liquid interface.

Generation of BHK-21 cell line expressing human ACE2 receptor. Lentiviral vectors encoding human ACE2 (GenBank NM_001371415.1) were produced as described previously72. BHK-21 cells were transduced with the ACE2-encoding lentivirus and selected in medium containing 200 μg/ml of hygromycin B. A pool of hygromycin-resistant cells, BHK-ACE2, was used in this study.

Generation of cell lines used for fusion assays. Retrovirus vectors were produced by transfecting HEK-293T cells with plasmid pQCXIP-GFP1-10 (Addgene #68715) or pQCXIP-BSR-GFP11 (Addgene #68716)42 and packaging vectors expressing MLV gal-pol and VSV-G using Lipofectamine 3000 (Invitrogen) according to manufacturer's instructions. Cell supernatants were harvested 24-48h post-transfection, pooled, clarified by centrifugation and filtered. One mL of each supernatant was used to transduce A549-Ace2-TMPRSS2 (AAT) cells72 in the presence of Polybrene (Merck). Two days post-transduction, the supernatant was replaced with selection medium (DMEM 10% FBS 1μg/mL puromycin) and cells incubated until complete death of the non-transduced control cells were observed. The resulting puromycin-resistant cells (termed AAT-GFP1-10 and AAT-BSR-GFP11) were used in fusion assays.

| | |
|---|---|
| Authentication | Not authenticated |
| Mycoplasma contamination | All cell lines were screened for Mycoplasma contamination |
| Commonly misidentified lines (See ICLAC register) | NA |

# Human research participants

Policy information about studies involving human research participants

| | |
|---|---|
| Population characteristics | Participants in the DOVE study were recipients of either 1, 2 or 3 doses of ChAdOx1, BNT162b2 or mRNA-1273 vaccines. Volunteers with immunosuppression and those who had previously been diagnosed with COVID-19 infection were excluded from the study. |
| Recruitment | Participants in the DOVE study were age but not sex-matched and gave written informed consent to participate in the study. |
| Ethics oversight | All participants in the DOVE study gave written informed consent to take part in the study which was approved by the North-West Liverpool Central Research Ethics Committee (REC reference 21/NW/0073). Residual nasopharyngeal swabs of patients infected with Omicron were collected with biorepository ethical approval (NHS Lothian reference 20/ES/0061). Derogated ethical approval for the use of demographic data for the EVADE study was granted by the NHS GG&C SafeHaven committee (GSH/21/IM/001). |

Note that full information on the approval of the study protocol must also be provided in the manuscript.

