## [Peer Review File · Nature Microbiology]

Peer Review Information

Journal: Nature Microbiology

Manuscript Title: SARS-CoV-2 Omicron is an immune escape variant with an altered cell entry pathway

Corresponding author name(s): Professor Emma Thomson

Reviewer Comments & Decisions:

Decision Letter, initial version:

11th January 2022

Dear Professor Palmarini,

Thank you very much for your enquiry about submitting your manuscript "The hyper-transmissible SARS-CoV-2 Omicron variant exhibits significant antigenic change, vaccine escape and a switch in cell entry mechanism" to Nature Microbiology. It certainly sounds interesting, and we would be happy to consider it for publication. However, I'm sure you'll understand that we cannot make a firm decision about whether to send the paper out to review until we have carefully read the full paper (and appropriate background literature).

In order to submit your complete manuscript to Nature Microbiology, please use the link below:

[Redacted]

If you have any questions, please feel free to contact me.

Yours sincerely,

[Redacted]

Decision Letter, first revision:

11th February 2022

Dear Emma,

thank you very much for your quick response and I'm glad that you will be able to clarify the aspect on the replicates as this was one of our main editorial concern. If you can also provide the suggested additional data and revise the manuscript within 2 or 3 weeks that will be very acceptable and certainly feasible for us. Please do let me know, however, if you expect any delays so that we can discuss how to proceed. Below are the instructions for revising the manuscript.

Thank you for your patience while your manuscript "SARS-CoV-2 Omicron is an immune escape variant with an altered cell entry pathway" was under peer-review at Nature Microbiology. It has now been seen by 2 referees, whose expertise and comments you will find at the of this email. You will see from their comments below that while they find your work of interest, some important points are raised. We are very interested in the possibility of publishing your study in Nature Microbiology, but would like to consider your response to these concerns in the form of a revised manuscript before we make a final decision on publication.

If you have not done so already please begin to revise your manuscript so that it conforms to our Article format instructions at <http://www.nature.com/nmicrobiol/info/final-submission/>

The usual length limit for a Nature Microbiology Article is six display items (figures or tables) and 3,000 words. We have some flexibility, and can allow a revised manuscript at 3,500 words, but please consider this a firm upper limit. There is a trade-off of ~250 words per display item, so if you need more space, you could move a Figure or Table to Supplementary Information.

Some reduction could be achieved by focusing any introductory material and moving it to the start of your opening 'bold' paragraph, whose function is to outline the background to your work, describe in a sentence your new observations, and explain your main conclusions. The discussion should also be limited. Methods should be described in a separate section following the discussion, we do not place a word limit on Methods.

Nature Microbiology titles should give a sense of the main new findings of a manuscript, and should not contain punctuation. Please keep in mind that we strongly discourage active verbs in titles, and that they should ideally fit within 90 characters each (including spaces).

Please include a data availability statement as a separate section after Methods but before references, under the heading "Data Availability". This section should inform readers about the availability of the data used to support the conclusions of your study. This information includes accession codes to public repositories (data banks for protein, DNA or RNA sequences, microarray, proteomics data etc...), references to source data published alongside the paper, unique identifiers such as URLs to data repository entries, or data set DOIs, and any other statement about data availability. At a minimum, you should include the following statement: "The data that support the findings of this study are available from the corresponding author upon request", mentioning any restrictions on availability. If DOIs are provided, we also strongly encourage including these in the Reference list (authors, title, publisher (repository name), identifier, year). For more guidance on how to write this section please see:

<http://www.nature.com/authors/policies/data/data-availability-statements-data-citations.pdf>

To improve the accessibility of your paper to readers from other research areas, please pay particular attention to the wording of the paper's opening bold paragraph, which serves both as an introduction and as a brief, non-technical summary in about 150 words. If, however, you require one or two extra sentences to explain your work clearly, please include them even if the paragraph is over-length as a result. The opening paragraph should not contain references. Because scientists from other sub-disciplines will be interested in your results and their implications, it is important to explain essential but specialised terms concisely. We suggest you show your summary paragraph to colleagues in other fields to uncover any problematic concepts.

If your paper is accepted for publication, we will edit your display items electronically so they conform to our house style and will reproduce clearly in print. If necessary, we will re-size figures to fit single

or double column width. If your figures contain several parts, the parts should form a neat rectangle when assembled. Choosing the right electronic format at this stage will speed up the processing of your paper and give the best possible results in print. We would like the figures to be supplied as vector files - EPS, PDF, AI or postscript (PS) file formats (not raster or bitmap files), preferably generated with vector-graphics software (Adobe Illustrator for example). Please try to ensure that all figures are non-flattened and fully editable. All images should be at least 300 dpi resolution (when figures are scaled to approximately the size that they are to be printed at) and in RGB colour format. Please do not submit Jpeg or flattened TIFF files. Please see also 'Guidelines for Electronic Submission of Figures' at the end of this letter for further detail.

Figure legends must provide a brief description of the figure and the symbols used, within 350 words, including definitions of any error bars employed in the figures.

Please include a statement before the acknowledgements naming the author to whom correspondence and requests for materials should be addressed.

Finally, we require authors to include a statement of their individual contributions to the paper -- such as experimental work, project planning, data analysis, etc. -- immediately after the acknowledgements. The statement should be short, and refer to authors by their initials. For details please see the Authorship section of our joint Editorial policies at http://www.nature.com/authors/editorial_policies/authorship.html

- * include a point-by-point response to any editorial suggestions and to our referees. Please include your response to the editorial suggestions in your cover letter, and please upload your response to the referees as a separate document.
- * ensure it complies with our format requirements for Letters as set out in our guide to authors at www.nature.com/nmicrobiol/info/gta/
- * state in a cover note the length of the text, methods and legends; the number of references;

number and estimated final size of figures and tables

* resubmit electronically if possible using the link below to access your home page:

[Redacted]

*This url links to your confidential homepage and associated information about manuscripts you may have submitted or be reviewing for us. If you wish to forward this e-mail to co-authors, please delete this link to your homepage first.

Please ensure that all correspondence is marked with your Nature Microbiology reference number in the subject line.

Nature Microbiology is committed to improving transparency in authorship. As part of our efforts in this direction, we are now requesting that all authors identified as 'corresponding author' on published papers create and link their Open Researcher and Contributor Identifier (ORCID) with their account on the Manuscript Tracking System (MTS), prior to acceptance. This applies to primary research papers only. ORCID helps the scientific community achieve unambiguous attribution of all scholarly contributions. You can create and link your ORCID from the home page of the MTS by clicking on 'Modify my Springer Nature account'. For more information please visit www.springernature.com/orcid.

We hope to receive your revised paper within three weeks. If you cannot send it within this time, please let us know.

Yours sincerely,

[Redacted]

Reviewer Expertise:

Referee #1: vaccine immune response

Referee #2: virus host cell interaction

Reviewers Comments:

Reviewer #1 (Remarks to the Author):

Willett et al., present a pertinent study on the impacts of VoC Omicron mutations on antibody evasion and cell entry pathways. Some minor clarifications in the text on the antibody specificity being OG Wuhan spike vs VoC antigen will help the reader, although they are implied.

Main comments:

1. Abstract could briefly describe vaccine sera source, i.e. AZ and mRNA vaccinated, and the approach to assess Omicron cell syncytia for context.
2. Introduction: Update BA.1 domination, as BA.2 sub lineage in Feb 2022 dominant, or proportion of sequences if available and state the time period, eg as of Feb 2022, due to rapid changes in the situation. Also update "including unpublished data made available as a press release from Pfizer" as peer reviewed papers now available (Cele Nature 2022, 10.1038/s41586-021-04387-1 and Cheng Nat Med 2022 10.1038/s41591-022-01704-7 among others)
3. To enable easier visualisation of Figure 1A, could the trimer subunits be shaded, and legend of RBD (ABC), NTD and FCS legend be added.
4. Page 4, Typo RBM?
5. Is there an update to the available sequences in the UK, Figure 1D?
6. Page 11, as the full dose for BNT162b2 is stated, what conc? Given mRNA-1273 (half dose; 50µg), describe the BNT162b2 ug at full dose.
7. Page 11, correct the vaccine antigen influenza Michigan H1 -> against the HA protein of A/Michigan/XXX/2021
8. Page 12, typo?, increased significantly against Omicron -> decreased significantly against Omicron ?
9. Clarify- "Infection-acquired immunity directed against other VOCs may be broader in nature"... broader antigen targets? As always that's IF you survive infection, therefore infection acquired immunity is not without risk and should be stated with a caveat of morbidity and mortality rather than being superior immune response.
10. The wording here is slightly confusing. As the vaccines contain the Wuhan Spike and the sVNT is for VoC, Omicron and Delta, "This level of protection was greater than that observed following two doses of OG [Wuhan Spike vaccine] for [VoC] Omicron but did not reach the levels attained by those who had never had natural [SARS-CoV-2] infection and had received third dose boosters for either Omicron or Delta" -> it sounds like the third dose is a Omicron/Delta, but again it is OG Wuhan- clarify the text.
11. Paragraph 2 and 3 of page 12, refer to figures within the text.
12. Page 13, live Omicron -> live Omicron virus. How was the virus isolated? Which cell line and passage number was used? Sequenced confirmed etc? ensure this is adequately described in the methods, and the strain lineage BA1? The methods say a TRMPS22 negative cell line, BHK-hACE2, cell line, not VAT successfully grew the initial virus from isolation, would this affect the entry results in Figure 4?
13. Page 13, formation n by Omicron – typo n?
14. Figure 4 EFG is a fantastic experiment and useful resource for cell pathwyas- was the downstream viral titer determined? i.e if only endosomal fusion is available is the viral titer lower? Relative infection and luciferase signals is shown. Is further data on viral titer available? If Omicron does not necessarily bind ACE2 but TRMPS22 for entry, should the Figure 4E show TRMPS22 alongside ACE2 rather than downstream of it?
15. Page 19, Discussion, 14% of CD8+ and 28% of CD4+ epitopes -> superscript + and add [T cell] epitopes.

Reviewer #2 (Remarks to the Author):

Willet et al. investigated the SARS-CoV-2 Omicron variant regarding immune evasion, cell entry

pathways and syncytium formation. They show that (i) Omicron displays a high level of immune escape with respect to serum from vaccinated people, (ii) enters cell predominantly via the endosomal (cathepsin B/L-dependent) route and (iii) is highly attenuated regarding the ability to drive cell-to-cell fusion (syncytium formation). For their experiments the authors used live virus, pseudoviruses and cell lines transfected to express the viral spike protein, which are established and widely-used methods for such analysis.

In the past weeks several studies on Omicron have been published, covering all key aspects of the present manuscript: Antibody escape of reduced neutralization sensitivity of the Omicron variant to sera from convalescent and vaccinated individuals have been already shown by multiple groups (e.g., Planas et al., 2021; Rössler et al., 2022; Hoffmann et al., 2022; Carreño et al., 2021; Muik et al., 2022; Cameron et al., 2021; ...). Similarly, reduced ability to cause cell-to-cell fusion (i.e., ability to drive syncytium formation) (e.g., Planas et al., 2021; Suzuki et al., 2022; Meng et al., 2022; Zhao et al., 2021) and altered cell entry pathways (e.g., Bojkova et al., 2022; Meng et al., 2022; Zhao et al., 2021) have been already reported by other groups.

In sum, the manuscript is well-written and presented data are clear. However, given that the key aspects of the manuscript have been already published in multiple studies by other groups, the overall novelty and impact of the present study is low and does not warrant publication in *Nature Microbiology*. Further, the fact that none of the experiments were replicated (as indicated in the reporting summary) and thus not confirmed in independent experiments is a major flaw of this study.

Specific point:

- In order to add novelty/impact and separate their study from published studies, the authors could elucidate which mutations in the Omicron spike protein are responsible for the characteristic changes in the cell entry pathway and ability to drive cell-to-cell fusion.

Minor points:

- How do the authors interpret their data on infection-acquired immunity (Fig. 3C) in the light of the findings by Rössler et al. (DOI: 10.1056/NEJMc2119236), which indicate that previous infections Alpha, Beta or Delta variants confers no to little protection against Omicron?
- Legend of Fig. 2, panel e: The legend states that mRNA-1273 booster samples are highlighted in green.

Author Rebuttal, first revision:

REPLY TO REVIEWER COMMENTS –OMICRON REVISION

9th March 2022

We thank the reviewers and editors for their time and for their comments and suggestions for improvement which are addressed below.

Reviewers Comments

Reviewer Expertise:

Referee #1: vaccine immune response

Referee #2: virus host cell interaction

Reviewer #1 (Remarks to the Author):

Willett et al., present a pertinent study on the impacts of VoC Omicron mutations on antibody evasion and cell entry pathways. Some minor clarifications in the text on the antibody specificity being OG Wuhan spike vs VoC antigen will help the reader, although they are implied.

Main comments:

1. Abstract could briefly describe vaccine sera source, i.e. AZ and mRNA vaccinated, and the approach to assess Omicron cell syncytia for context.

The abstract has been updated as suggested

2. Introduction: Update BA.1 domination, as BA.2 sub lineage in Feb 2022 dominant, or proportion of sequences if available and state the time period, eg as of Feb 2022, due to rapid changes in the situation. Also update “including unpublished data made available as a press release from Pfizer” as peer reviewed papers now available (Cele Nature 2022, 10.1038/s41586-021-04387-1 and Cheng Nat Med 2022 10.1038/s41591-022-01704-7 among others)

Updated as suggested

3. To enable easier visualisation of Figure 1A, could the trimer subunits be shaded, and legend of RBD (ABC), NTD and FCS legend be added.

Done

4. Page 4, Typo RBM?

Corrected

5. Is there an update to the available sequences in the UK, Figure 1D?

Updated 1D

6. Page 11, as the full dose for BNT162b2 is stated, what conc? Given mRNA-1273 (half dose; 50µg), describe the BNT162b2 ug at full dose.

Clarified as requested

7. Page 11, correct the vaccine antigen influenza Michigan H1 -> against the HA protein of A/Michigan/XXX/2021

Changed as advised to appropriate nomenclature

8. Page 12, typo?, increased significantly against Omicron -> decreased significantly against Omicron ?

Updated text to clarify

9. Clarify- “Infection-acquired immunity directed against other VOCs may be broader in nature”... broader antigen targets? As always that’s IF you survive infection, therefore infection acquired immunity is not without risk and should be stated with a caveat of morbidity and mortality rather than being superior immune response.

We adjusted the text here to specify “natural infection” rather than “other VOCs”. Also, we have added text to emphasise this important point about surviving infection in the same paragraph.

10. The wording here is slightly confusing. As the vaccines contain the Wuhan Spike and the sVNT is for VoC, Omicron and Delta, “This level of protection was greater than that observed following two doses of OG [Wuhan Spike vaccine] for [VoC] Omicron but did not reach the levels attained by those who had never had natural [SARS-CoV-2] infection and had received third dose boosters for either Omicron or Delta” -> it sounds like the third dose is a Omicron/Delta, but again it is OG Wuhan- clarify the text.

Text clarified

11. Paragraph 2 and 3 of page 12, refer to figures within the text.

Figure references added (these have changed slightly following the addition of extra data as described later)

12. Page 13, live Omicron -> live Omicron virus. How was the virus isolated? Which cell line and passage number was used? Sequenced confirmed etc? ensure this is adequately described in the

methods, and the strain lineage BA1? The methods say a TRMPS22 negative cell line, BHK-hACE2, cell line, not VAT successfully grew the initial virus from isolation, would this affect the entry results in Figure 4?

As requested, we have expanded within the Materials and Methods, the “Virus isolation from clinical samples” section. Indeed, we sequenced the isolated viruses and deposited the sequence in GISAID as is now indicated in the text. We have also briefly described the methods followed and confirmed that sample 205 is identified as belonging to the Omicron BA.1 lineage by Pangolin.

With regards to the comment made on BHK-hACE2 cells, these cells do not overexpress TMPRSS2 but they are not TMPRSS2 negative. We also don’t expect that the initial passage (p0) in BHK-hACE2 could have changed the phenotype of the virus with regards to entry as we have sequenced sample 205 as passage 1 and 2 (P1) in cells other than BHK-hACE2 and the sequence obtained has no significant spike mutations from the Omicron sequences circulating at that time.

Importantly, our live virus-based entry assays are in good agreement with plasmid-based spike experiments (cell-cell fusion and pseudotypes), in which spike is genetically static. This would argue against any artefact based on the selection of adapted viruses with altered entry characteristics during the initial isolation in BHK cells.

13. Page 13, formation n by Omicron – typo n?

Corrected

14. Figure 4 EFG is a fantastic experiment and useful resource for cell pathways- was the downstream viral titer determined? i.e if only endosomal fusion is available is the viral titer lower?

These experiments were conducted with non-replicating pseudotypes bearing SARS-CoV-2 spike and encoding a luciferase reporter gene. Consequently, expression of luciferase provides a readout of spike-mediated entry. However, being replication-deficient there are no subsequent rounds of infection to generate viral titres. Nonetheless, our revised manuscript contains extensive new experiments using live Omicron which complement the pseudotype experiments and suggest that Omicron tissue preference is determined by the relative availabilities of either cell-surface or endosomal fusion.

If Omicron does not necessarily bind ACE2 but TRMPS22 for entry, should the Figure 4E show TRMPS22 alongside ACE2 rather than downstream of it?

There is currently no evidence that Omicron has dispensed with ACE-2 as a receptor. Moreover, the primary role of TMPRSS2 is to proteolytically activate spike, a process that is thought to occur after initial interactions with ACE-2; therefore, TMPRSS2 should sit downstream of ACE2 in Fig. 4E. In the case

of Omicron, our, and others', data suggests that it can use an alternative, endosomal, protease instead of TMPRSS2.

15. Page 19, Discussion, 14% of CD8+ and 28% of CD4+ epitopes -> superscript + and add [T cell] epitopes.

Corrected as suggested. This section has also been expanded now to include T cell data.

Reviewer #2 (Remarks to the Author):

Willet et al. investigated the SARS-CoV-2 Omicron variant regarding immune evasion, cell entry pathways and syncytium formation. They show that (i) Omicron displays a high level of immune escape with respect to serum from vaccinated people, (ii) enters cell predominantly via the endosomal (cathepsin B/L-dependent) route and (iii) is highly attenuated regarding the ability to drive cell-to-cell fusion (syncytium formation). For their experiments the authors used live virus, pseudoviruses and cell lines transfected to express the viral spike protein, which are established and widely-used methods for such analysis.

In the past weeks several studies on Omicron have been published, covering all key aspects of the present manuscript: Antibody escape of reduced neutralization sensitivity of the Omicron variant to sera from convalescent and vaccinated individuals have been already show by multiple groups (e.g., Planas et al., 2021; Rössler et al., 2022; Hoffmann et al., 2022; Carreño et al., 2021; Muik et al., 2022; Cameroni et al, 2021; ...). Similarly, reduced ability to cause cell-to-cell fusion (i.e., ability to drive syncytium formation) (e.g., Planas et al., 2021; Suzuki et al., 2022; Meng et al., 2022; Zhao et al., 2021) and altered cell entry pathways (e.g., Bojkova et al., 2022; Meng et al., 2022; Zhao et al., 2021) have been already reported by other groups.

In sum, the manuscript is well-written and presented data are clear. However, given that the key aspects of the manuscript have been already published in multiple studies by other groups, the overall novelty and impact of the present study is low and does not warrant publication in Nature Microbiology.

We recognise the point about novelty but this is simply an issue around timing of submission and review. We were in fact the first to describe the switch in entry preference as evidenced by our preprint from late December 2021 that has been extensively cited

<https://www.medrxiv.org/content/10.1101/2022.01.03.21268111v1.article-metrics>

Nevertheless, we have added substantial new data to add novelty to our findings.

Further, the fact that none of the experiments were replicated (as indicated in the reporting summary) and thus not confirmed in independent experiments is a major flaw of this study.

This is an error in our part in the reporting summary which on second reading was indeed misleading – however we can provide assurance that all the *in vitro* experiments were all repeated/carried out with replicates. The comment in the reporting summary referred only to the population-based vaccine effectiveness data which couldn't be replicated at the time because of the timing of the data available in the population. In any case, we have now had the opportunity to look at vaccine effectiveness again with much larger numbers and refined methodology and our estimates are similar but with much higher confidence intervals. As the reviewer points out, much of the initial data were also in keeping with other studies that were published while our paper was under review, providing further reassurance. We have also now substantially updated our findings to provide novelty, as suggested by the reviewer, as we anticipated a comment along these lines and already had these experiments underway.

Specific point:

- In order to add novelty/impact and separate their study from published studies, the authors could elucidate which mutations in the Omicron spike protein are responsible for the characteristic changes in the cell entry pathway and ability to drive cell-to-cell fusion.

To understand the determinants of the Omicron spike phenotype we performed domain swaps with the ancestral Wuhan spike, these experiments yielded important and novel findings. Whilst efficient endosomal entry mapped to the S2 portion of Omicron spike (with minor contributions by the NTD), the cell-cell fusion defect (and associated changes in proteolysis) were determined by the RBD of Omicron spike. This suggests a complex interplay of domains bearing cooperative and, likely, compensatory mutations. The ability of Omicron RBD to regulate proteolytic processing at the S1/S2 junction was not expected; this finding will be of great interest to the field.

Aside from these insights, we have also added further virological characterisation of Omicron in primary human nasal cells, further exploration of the cell-fusion phenotype and evaluation of Omicron BA.2 spike. These experiments have added further novelty, mechanistic depth and physiological relevance to our existing experiments. Finally, we have added data on neutralisation of BA.1, BA.1.1 and BA.2 by vaccine sera and monoclonal antibodies and for completeness have added T cell ELISpot data to investigate immunity from this arm of the adaptive immune response.

Minor points:

- How do the authors interpret their data on infection-acquired immunity (Fig. 3C) in the light of the findings by Rössler et al. (DOI: 10.1056/NEJMc2119236), which indicate that previous infections Alpha, Beta or Delta variants confers no to little protection against Omicron?

For the vaccine effectiveness population data, we have now expanded our analysis to look at the impact of previous infection plus vaccination versus vaccination alone on test positivity (Figure 4c). Our findings are in keeping with the live virus neutralisation data shown by Rössler *et al* as follows:

1. “Serum samples from vaccinated persons neutralized the omicron variant to a much lesser extent than any other variant analyzed”.

In keeping with this neutralisation data, we have also shown (with the pseudovirus system) markedly reduced neutralisation. We have also shown a marked drop in vaccine effectiveness in a population of 1.2 million people.

2. “Serum samples that were obtained from convalescent participants largely did not neutralize the omicron variant”

We looked at unvaccinated convalescent patients (Figure 4c) and found very low protection against reinfection in this group (16.3%). This protection may be mediated by B or T-cell responses (we have now shown in this paper that T cell responses are relatively conserved against Omicron).

We have also expanded on the Rössler study by investigating the effect of three doses of vaccine on neutralisation and vaccine effectiveness. Booster vaccine (even with a Wuhan-based vaccine) is remarkably effective, especially in those who have had full vaccination and a previous infection.

- Legend of Fig. 2, panel e: The legend states that mRNA-1273 booster samples are highlighted in green.

Thank you -this is now corrected

Decision Letter, second revision:

: Our ref: NMICROBIOL-21123153B

29th March 2022

Dear Emma,

Thank you for submitting your revised manuscript "SARS-CoV-2 Omicron is an immune escape variant with an altered cell entry pathway" (NMICROBIOL-21123153B). It has now been seen by the original referees and their comments are below. The reviewers find that the paper has improved in revision, and therefore we'll be happy in principle to publish it in Nature Microbiology, pending minor revisions to satisfy the referees' final requests and to comply with our editorial and formatting guidelines. Please note that you won't be required to add additional data (as suggested by reviewer #2).

We are now performing detailed checks on your paper and will send you a checklist detailing our

editorial and formatting requirements in about a week. Please do not upload the final materials and make any revisions until you receive this additional information from us.

Thank you again for your interest in Nature Microbiology. Please do not hesitate to contact me if you have any questions.

Sincerely,

[Redacted]

Reviewer #1 (Remarks to the Author):

The authors have clarified the issues raised by both reviewers for a stronger manuscript. They have performed extensive experiments since the first version of the manuscript with BA2, third dose immune serum and Spike domain specific mapping for binding- this is a significant and important study for understanding the impact on viral entry and immune evasion that makes Omicron the most significant VoC to date- commendations to the authors.

Minor comments

1. Figure 1 B mutant label text is difficult to read- can the font or resolution be increased.
2. Discussion: 14% of CD8+ and 28% of CD4+ epitopes -> not updated to superscript and add T cell, i.e. CD4+ T cell epitopes.
3. When referring to Omicron BA1 and/or BA2 should also be stipulated (Fig 4/5 as Omicron (BA1)).

Reviewer #2 (Remarks to the Author):

The authors have appropriately addressed the questions raised by this reviewer. The newly added data significantly increase the novelty/quality of the manuscript. Particularly, the new data on domain swap between ancestral and omicron spike proteins are indeed exciting findings that will be of great interest to the field. For this reviewer it would be interesting to know whether the authors also tested entry of pseudoviruses bearing chimeric spike proteins for Calu-3 cells (route 2)? It would be nice to include such data (if available). However, absence of such data does not compromise my enthusiasm for this study and I can recommend it for publication.

Decision Letter, final checks:

Our ref: NMICROBIOL-21123153B

1st April 2022

Dear Emma,

Thank you for your patience as we've prepared the guidelines for final submission of your Nature Microbiology manuscript, "SARS-CoV-2 Omicron is an immune escape variant with an altered cell entry pathway" (NMICROBIOL-21123153B). Please carefully follow the step-by-step instructions provided in the attached file, and add a response in each row of the table to indicate the changes that

you have made. Please also check and comment on any additional marked-up edits we have proposed within the text. Ensuring that each point is addressed will help to ensure that your revised manuscript can be swiftly handed over to our production team. I would also like to point out again that we do not require you to add any new data in response to reviewer #2 also to avoid any delays in processing the paper as any new data would have to be assessed by the reviewers again. Please let me know if you have any questions!

In recognition of the time and expertise our reviewers provide to Nature Microbiology's editorial process, we would like to formally acknowledge their contribution to the external peer review of your manuscript entitled "SARS-CoV-2 Omicron is an immune escape variant with an altered cell entry pathway". For those reviewers who give their assent, we will be publishing their names alongside the published article.

Nature Microbiology offers a Transparent Peer Review option for new original research manuscripts submitted after December 1st, 2019. As part of this initiative, we encourage our authors to support increased transparency into the peer review process by agreeing to have the reviewer comments, author rebuttal letters, and editorial decision letters published as a Supplementary item. When you submit your final files please clearly state in your cover letter whether or not you would like to participate in this initiative. Please note that failure to state your preference will result in delays in accepting your manuscript for publication.

Cover suggestions

As you prepare your final files we encourage you to consider whether you have any images or illustrations that may be appropriate for use on the cover of Nature Microbiology.

Please submit your suggestions, clearly labeled, along with your final files. We'll be in touch if more

information is needed.

Nature Microbiology has now transitioned to a unified Rights Collection system which will allow our Author Services team to quickly and easily collect the rights and permissions required to publish your work. Approximately 10 days after your paper is formally accepted, you will receive an email in providing you with a link to complete the grant of rights. If your paper is eligible for Open Access, our Author Services team will also be in touch regarding any additional information that may be required to arrange payment for your article.

Please note that *Nature Microbiology* is a Transformative Journal (TJ). Authors may publish their research with us through the traditional subscription access route or make their paper immediately open access through payment of an article-processing charge (APC). Authors will not be required to make a final decision about access to their article until it has been accepted. [Find out more about Transformative Journals](https://www.springernature.com/gp/open-research/transformative-journals)

Authors may need to take specific actions to achieve [compliance with funder and institutional open access mandates](https://www.springernature.com/gp/open-research/funding/policy-compliance-faqs). If your research is supported by a funder that requires immediate open access (e.g. according to [Plan S principles](https://www.springernature.com/gp/open-research/plan-s-compliance)) then you should select the gold OA route, and we will direct you to the compliant route where possible. For authors selecting the subscription publication route, the journal's standard licensing terms will need to be accepted, including [self-archiving policies](https://www.springernature.com/gp/open-research/policies/journal-policies). Those licensing terms will supersede any other terms that the author or any third party may assert apply to any version of the manuscript.

Please use the following link for uploading these materials:
[Redacted]

Best regards,

[Redacted]

Reviewer #1:

Remarks to the Author:

The authors have clarified the issues raised by both reviewers for a stronger manuscript. They have performed extensive experiments since the first version of the manuscript with BA2, third dose immune serum and Spike domain specific mapping for binding- this is a significant and important study for understanding the impact on viral entry and immune evasion that makes Omicron the most significant VoC to date- commendations to the authors.

Minor comments

1. Figure 1 B mutant label text is difficult to read- can the font or resolution be increased.
2. Discussion: 14% of CD8+ and 28% of CD4+ epitopes -> not updated to superscript and add T cell, i.e. CD4+ T cell epitopes.
3. When referring to Omicron BA1 and/or BA2 should also be stipulated (Fig 4/5 as Omicron (BA1)).

Reviewer #2:

Remarks to the Author:

The authors have appropriately addressed the questions raised by this reviewer. The newly added data significantly increase the novelty/quality of the manuscript. Particularly, the new data on domain swap between ancestral and omicron spike proteins are indeed exciting findings that will be of great interest to the field. For this reviewer it would be interesting to know whether the authors also tested entry of pseudoviruses bearing chimeric spike proteins for Calu-3 cells (route 2)? It would be nice to include such data (if available). However, absence of such data does not compromise my enthusiasm for this study and I can recommend it for publication.

Final Decision Letter:

Dear Emma,

I am pleased to accept your Article "SARS-CoV-2 Omicron is an immune escape variant with an altered cell entry pathway" for publication in Nature Microbiology. Thank you for having chosen to submit your work to us and many congratulations.

Due to the importance of these deadlines, we ask you to please let us know now whether you will be difficult to contact over the next month. If this is the case, we ask you to provide us with the contact information (email, phone and fax) of someone who will be able to check the proofs on your behalf, and who will be available to address any last-minute problems.

Acceptance of your manuscript is conditional on all authors' agreement with our publication policies (see <https://www.nature.com/nmicrobiol/editorial-policies>). In particular your manuscript must not be published elsewhere and there must be no announcement of the work to any media outlet until the publication date (the day on which it is uploaded onto our website).

Please note that *Nature Microbiology* is a Transformative Journal (TJ). Authors may publish their research with us through the traditional subscription access route or make their paper immediately open access through payment of an article-processing charge (APC). Authors will not be required to make a final decision about access to their article until it has been accepted. [Find out more about Transformative Journals](https://www.springernature.com/gp/open-research/transformative-journals)

Authors may need to take specific actions to achieve [compliance](https://www.springernature.com/gp/open-research/funding/policy-compliance-faqs) with funder and institutional open access mandates. If your research is supported by a funder that requires immediate open access (e.g. according to [Plan S principles](https://www.springernature.com/gp/open-research/plan-s-compliance)) then you should select the gold OA route, and we will direct you to the compliant route where possible. For authors selecting the subscription publication route, the journal's standard licensing terms will need to be accepted, including [self-archiving policies](https://www.nature.com/nature-portfolio/editorial-policies/self-archiving-and-license-to-publish). Those licensing terms will supersede any other terms that the author or any third party may assert apply to any version of the manuscript.
